# Identification of stratospheric disturbance information in China based on round-trip intelligent sounding system

Yang He[1], Xiaoqian Zhu[1], Zheng Sheng*[1], Mingyuan He[1]

[1] College of Meteorology and Oceanography, National University of Defense Technology, Changsha, 410073, China

*Correspondence to*: Zheng Sheng (19994035@sina.com)

**Abstract.** Assessing the role of physical processes in the stratosphere under climate change has been one of the hottest topics over the past few decades. However, due to the limitation of detection technique, the stratospheric disturbance information from in situ observation is still relatively scarce. The round-trip intelligent sounding system (RTISS) is a new detection technology developed in recent years, which can capture atmospheric fine structure information of the troposphere and stratosphere through the three-stage (rising, flat-floating, and falling) detection. Based on the structure function and singular measure, we quantify the stratospheric small-scale gravity wave (SGW) over China by Hurst parameter and intermittency parameter, and discuss its relationship with inertia-gravity wave (IGW). The results show that the enhancement of the SGWs in the stratosphere is accompanied by the weakening of the IGWs below, which is related to the Kelvin-Helmholtz instability (KHI), and is conducive to the transport of ozone to higher altitudes from lower stratosphere. The parameter space (H1, C1) shows sufficient potential in the analysis of stratospheric disturbances and their role in material transport and energy transfer.

## 1 Introduction

Gravity waves (GWs) are waves generated by gravity and are widespread in the earth's atmosphere. GWs are excited by wave sources in the troposphere, including topography, convection, and wind shear, etc, and propagate from the troposphere to the stratosphere and higher altitudes (Alexander et al., 2010; Fritts and Alexander, 2003, 2012). During upward propagation of GWs, due to the decrease of atmospheric density and the increase of wave amplitudes (Fritts and Alexander, 2003; Mohankumar, 2008), the influence of GWs on the surrounding atmosphere is increasingly important. This effect is mainly caused by the instability of GWs with increasing amplitude, or the breaking of GWs when they encounter the "critical layer", thus changing the circulation and structure of the atmosphere by dissipating energy and momentum (Allen and Vincent, 1995; Hertzog et al., 2012).

In order to improve the simulation of the main average characteristics of the atmosphere by numerical weather prediction (NWP) and general circulation models (GCMs), it is necessary to describe important physical processes in the atmosphere more accurately and efficiently (Kim et al., 2003). Part of the GWs have relatively small scales and cannot be resolved in models with relatively rough resolution, so it is necessary to use a parameterization to describe the influence and interaction

of GWs on larger-scale dynamic process. The GW parameterization is now a key component of almost all large-scale atmospheric models. However, due to the lack of observational constraints and insufficient understanding of the mechanism, it also restricts the prediction accuracy and simulation ability of the models (Plougonven et al., 2020).

Assessing the role of stratospheric physical processes under climate change is one of the hottest topics in the past few decades (SPARC, 2022; Tian et al., 2023). GWs, as one of the important physical processes in the stratosphere, has been

extensively studied, based on radiosonde (Kinoshita et al., 2019; Moffat-Griffin et al., 2013), rocket (Eckermann et al., 1995), radar (Alexander et al., 2017; Huang et al., 2017), remote sensing (Wright et al., 2016; Guo et al., 2021) and other detection methods. Limited by the detection technology, relatively little research has been carried out on the fine structure of the stratospheric atmosphere. Aircraft observation can only be used for specific design tasks (Zhang et al., 2015), with little continuous data accumulation. Super-pressure balloons can provide stratospheric GW field information on particular zonal

circles with long-duration observation (Alexander et al., 2021; Hertzog et al., 2008), though it is not currently applicable to local areas within countries.

At present, the stratospheric disturbance information in the horizontal direction is still relatively scarce in China, and the introduction of the flat-floating information can help to improve the forecasting effect of the models and deepen the understanding of stratospheric dynamic processes (Laroche et al., 2013; Cohn et al., 2013). The round-trip intelligent sounding

system (RTISS) is a new detection technology developed in recent years (Cao et al., 2019), which can capture atmospheric fine structure information of the troposphere and stratosphere through the three-stage (rising, flat-floating, and falling) detection. That is, the outer balloon carries the radiosonde for ascending detection, and the inner balloon continues to carry the radiosonde for stratospheric detection after the outer balloon explodes, and the radiosonde is carried by the parachute for descending detection after the flat-floating is over. For the first time, this paper shows a relatively complete analysis of

atmospheric disturbance information in the horizontal direction of the stratosphere in China through RTISS, and provides an innovative result for the evaluation of physical processes in the stratosphere.

## 2 Observation from RTISS

### 2.1 Introduction to experimental data

Data used in the paper are from experimental project of round-trip intelligent sounding system (RTISS), covering six sites

including Yichang (YC), Wuhan (WH), Anqing (AQ), Changsha (CS), Nanchang (NC), and Ganzhou (GZ) in China. RTISS can realize the three-stage detection including "rising, flat-floating, and falling", which has become an important source for the analysis of atmospheric disturbance information in the horizontal direction of the stratosphere (Cao et al., 2019; He et al., 2022). The release time span is from June 1 to July 10 (summer), and from October 13 to November 18 (autumn) in 2018. There are 245 detections in autumn (34 in AQ, 34 in GZ, 46 in NC, 43 in WH, 47 in YC, and 41 in CS) and 245 detections in

summer (40 in AQ, 48 in GZ, 43 in NC, 44 in WH, 50 in YC, and 20 in CS).

The details of the observation experiment are shown in Figure 1. The flat-floating height covers the range of 18–32 km, mainly concentrated in 26–30 km (Figure 1a), and the variation of height over time during the entire detection process is shown in Fig. A1. Six sites are all located in southeast China (Figure 1b). The balloon trajectories can directly reflect the stratospheric wind field characteristics over the corresponding sites (Figure 1 c–h). In summer, the stratosphere is mainly dominated by easterly winds, with relatively stable circulation (more consistent trajectories), while in autumn, circulation changes more frequently (more divergent trajectories).

In order to explore the correlation between RTISS data and atmospheric composition, we obtained ozone and potential vorticity from ERA5 reanalysis data (0.25°×0.25°). The release time of flat-floating detection is divided into two periods, morning and evening. The release is done approximately at 23UTC (7:00 Beijing time) and 11UTC (19:00 Beijing time). Taking into account the rise time of nearly 1–1.5 hour, it arrives upward at approximately 00 UTC and 12UTC for flat-floating detection. Therefore, the 00UTC and 12UTC data provided by ERA5 can be well combined with the observation results of RTISS in the flat-floating stage for analysis.

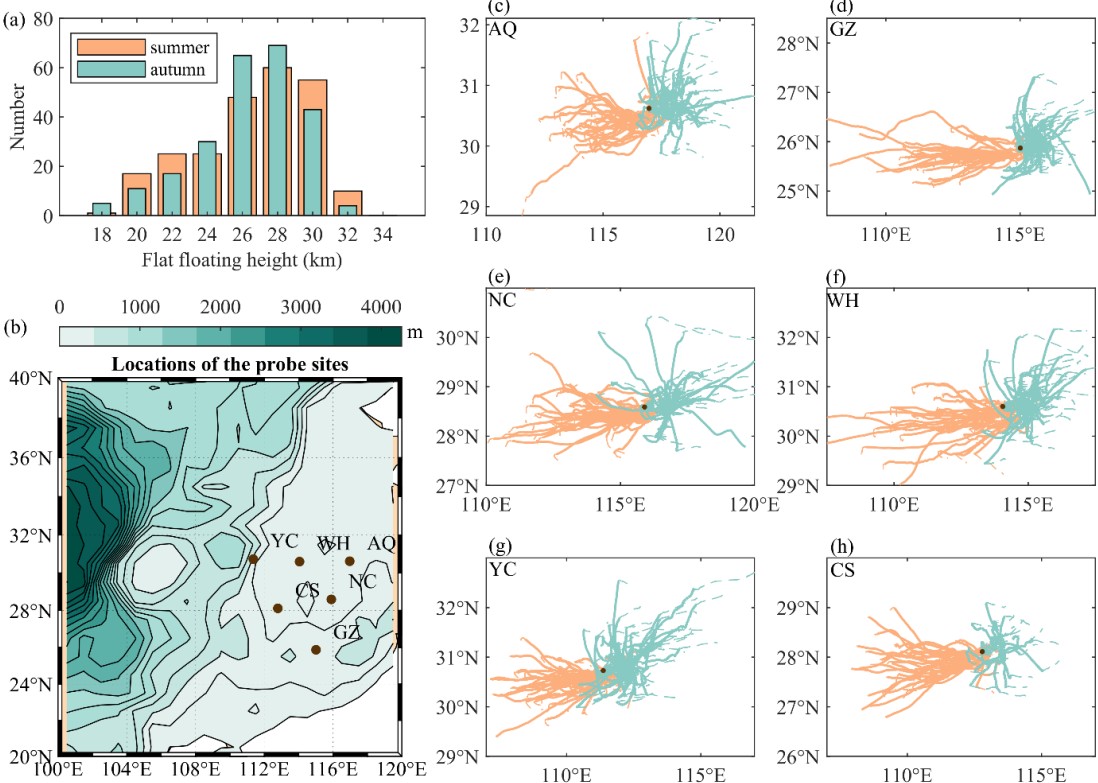

**Figure 1. (a) Histogram of flat-floating height, (b) A topographic map of the RTISS release sites and nearby areas, and the trajectories of RTISS over (c) Anqing, (d) Ganzhou, (e) Nanchang, (f) Wuhan, (g) Yichang, and (h) Changsha. The black dots represent the release sites, the dashed lines represent trajectories during rising and falling stages, and the solid lines represent trajectories during flat-floating stage. In order to better compare the results of different sites, the axis of the c–h subgraph is unified into the same geographic width (10°×4°)**

**2.2 Detection principle and quality control**

RTISS aims to maintain a relatively low cost while achieving encrypted observations several hours apart in the vertical direction (several hours between the end of the detection in the rising stage and the beginning of the detection in the falling stage), as well as continuous high-frequency observations (1s) for several hours at a specific altitude (flat-floating), to capture the atmospheric fine structure information from the troposphere to the stratosphere, including wind field, temperature, air

pressure, and relative humidity (RH). The sounding instrument carries the Beidou navigation system and the meteorological sensor. The Beidou navigation system provides positioning information (longitude, latitude, altitude) that can be used to calculate the horizontal wind field. The uncertainty of wind speed is 2 m/s during rising stage and 4 m/s during flat-floating stage. The sensor module can be used to obtain temperature, RH, and air pressure, which consists of three parts: (1) a negative temperature coefficient (NTC) thermistor sensor for temperature measurement, with an uncertainty of 0.8 K during rising stage

and 2.8 K during flat-floating stage; (2) a piezoresistive sensor for air pressure measurement, with an uncertainty of 1 hPa during the rising stage and flat-floating stage; and (3) a humidity-sensitive capacitance sensor, with an uncertainty of 10% RH during the rising stage, while it is ignored during flat-floating stage with poor data quality. The uncontrolled, high-velocity descent through parachute during falling stage may influence measurement quality with a strong pendulum motion (Jorge et al., 2021), so we do not consider the data in this stage.

The three-stage detection process by RITSS is described in Figure 2. In the rising stage, the two-balloon method (an inner balloon inside an outer balloon) is used to carry the radiosonde up and make real-time measurements. When a predetermined height is reached, the outer balloon is exploded, at that time, the buoyancy of the inner balloon is just equal to the gravity of the carried instrument, and it drifts with the wind at the predetermined height with a quasi-horizontal movement. When the balloon floats for several hours to reach the predetermined area, the radiosonde and the inner balloon are separated by a fuse

device, then the parachute above the instrument opens, carrying the instrument descends.

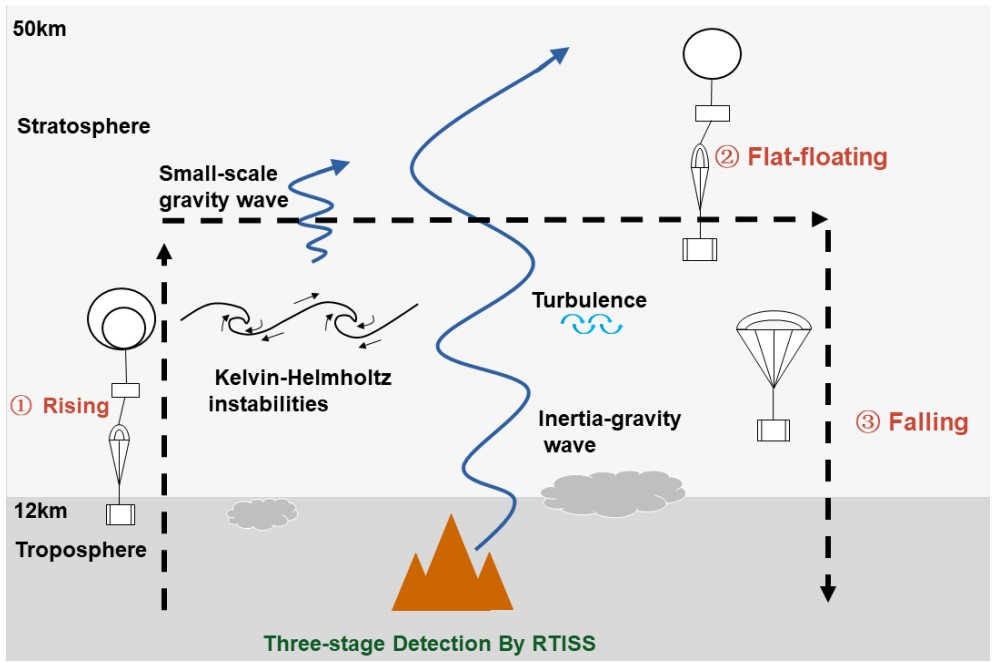

**Figure 2. The three-stage detection process by RITSS**

The detection system has different working principles in the three stages, and the specific dynamic process can be referred to the previous work (Cao et al., 2019). It should be noted that the RTISS uses the zero-pressure balloon to meet the needs of low-cost business observation, which is different from the super-pressure balloon (Hertzog et al., 2008). For the zero-pressure balloon, the bottom exhaust pipe makes the pressure difference between inside and outside the balloon basically zero, the flight time is short (several hours). While for the super-pressure balloon, the sphere is closed, and the volume of the sphere is basically unchanged, with longer flight time (several weeks).

It is known that balloon payload can have a pendulum motion (Andreas et al., 2016), and we have selected the appropriate smooth fitting interval to eliminate its effect. An integer multiple of the swing period is used as the smooth fitting interval, and the symmetry of the swing is used to compensate for the swing deviation. Using the average smoothed position coordinates, the first derivative is obtained by linear fitting to obtain the speed, and the second derivative is obtained by quartic fitting to obtain the acceleration. Then wind speed and wind direction can be obtained after that. It should be noted that different smoothing points may cause some difference in the quantization results of gravity waves. However, if all data sets are smoothed in the same way, the internal comparison will not be affected by this.

The variation of the height during the whole process of RTISS over time is shown in Figure A1. In order to ensure the premise of approximate constant height, we need to sift through all the flat-floating data, and only data sets with a long enough flat-floating time (longer than 3–4 hours) and relatively good flat-floating quality (the difference between the maximum and minimum height is within several hundred meters) are selected. It should be noted that, after the burst of the outer balloon, the platform adjusts to its equilibrium level a few hundred meters below the burst altitude (Figure A1), thus the initial segment

after the burst of the outer balloon is also discarded. Along the measured points, the flat-floating distance is usually tens of kilometers to hundreds of kilometers (in the same height plane), and the fluctuation of several hundred meters in the vertical direction can still be approximated as quasi-horizontal movement. The original data is tested for horizontal consistency, and then re-interpolated to a uniform spatial interval after the outlying and missing values are removed.

## 3 Analysis method

### 3.1 Third-order structure function

In order to effectively identify the atmospheric disturbance information obtained by RTISS, we consider combining the results from the rising and flat-floating stages for the analysis, while the falling stage is not included due to the relatively poor data quality. We assume that RTISS can capture the same weather system during the rising and flat-floating stages due to the continuous observation in space and time. The observation results in the horizontal and vertical directions can just complement each other, which is currently impossible for other single observations.

We use the third-order structure function $S_3(r)$ to identify GWs and turbulence. This method was earlier used in aircraft observation data (Cho and Lindborg, 2001). At the tail of the $S_3(r)$ (turbulence subrange), the $r$ slope represents the occurrence of turbulence, while in the larger scales (GW subrange) of $S_3(r)$, the $r^2$ and $r^3$ slope represent the unstable and stable GWs, respectively (Lu and Koch, 2008; He et al., 2022). The calculation is as follows (Cho and Lindborg, 2001; Lindborg, 1999):`

$$S_3(r) = \langle[\delta u_L(r)]^3\rangle + 2\langle\delta u_L(r)[\delta u_T(r)]^2\rangle = -\frac{4}{3}Er. \tag{1}$$

Among them, $\langle.\rangle$ is the ensemble average, $r$ is the separation distance, and $E$ is the energy dissipation rate. The balloon trajectory during flat-floating stage is not a straight line, so we decompose it into the zonal and meridional directions, and take the direction of the longer projection distance as the separation distance direction. Separation distance can be determined as $r = l \times 2^n$, for integers $n = 0, 1 \dots, N$, where $l$ is the average step along the separation distance direction, and $N$ is limited by the maximum data length $L$ (in the current data $N = 13$ or $N = 14$). The directions parallel to and perpendicular to the separation distance is represented by L and T, respectively. The raw data is uniformly interpolated to the average step along the separation distance direction.

### 3.2 Hurst index and intermittent parameter

Similar to Eq. (1), the multi-order structure function is defined as:

$$S_q(r) = \langle|u_L(x + r) - u_L(x)|^q\rangle = \langle|\delta u_L(r)|^q\rangle, \tag{2}$$

where $0 \ll x \ll L - r$. It should be noted that Eq. (1) is used to identify the state of the GWs and the energy cascade direction, while Eq. (2) is used to calculate the subsequent disturbance parameters, consistent with previous studies (Lu and

Koch, 2008; Marshak et al., 1997). Assuming that this process is scale-invariant and self-similar, $S_q(r)$ can be scaled to (Lu and Koch, 2008):

$$S_q(r) = C_q r^{\zeta(q)}, q \geq 0, \tag{3}$$

where $C_q$ is a constant and $\zeta(q)$ is a function of order $q$. From this we can define a monotone, non-increasing function
(Marshak et al., 1997):

$$H(q) = \frac{\zeta(q)}{q}. \tag{4}$$

Here we define H1=$H(1)$ as the Hurst index, which can measures the roughness (nonstationarity) of the signal in data, with a value between 0–1 (Marshak et al., 1997). The larger the H1, the smoother the data sequence and the fewer wave packets superimposed on it, and vice versa.

Statistical analysis called singularity measurement can be used to reflect the intermittency of the data sequence (Marshak et al., 1997), a non-negative normalized η-scale gradient field is defined by a second-order structure function (Lu and Koch, 2008):

$$\varepsilon(\eta; x) = \frac{|\delta u_L(x,\eta)|^2}{\langle |\delta u_L(x,\eta)|^2 \rangle}, \eta \leq x \leq L - r, \tag{5}$$

where $L$ is the maximum length of the data, and $\eta = 4l$ is four times the Nyquist wavelength. The measurements at
different separation distances r can be expressed by the results of spatial averaging:

$$\varepsilon(r; x) = \frac{1}{r} \int_x^{x+r} \varepsilon(\eta; x') \, dx', \ \eta \leq x \leq L - r. \tag{6}$$

The self-similarity of fluctuations makes the q-order measurement expressed as:

$$\langle \varepsilon(r; x)^q \rangle = \langle \varepsilon(r)^q \rangle \propto r^{-K(q)}, q \geq 0. \tag{7}$$

By linearly fitting the $\varepsilon(r)$ curves of different orders $q$, the $K(q)$ curve can be obtained. Then the generalized dimension
is introduced:

$$D(q) = 1 - \frac{K(q)}{q-1}. \tag{8}$$

The intermittent nature of fluctuations can be expressed as:

$$C1 = 1 - D(1) = 1 - \lim_{q \to 1} D(q) = \lim_{q \to 1} \frac{K(q)}{q-1} = K'(1). \tag{9}$$

C1 is an intermittent parameter with a value between 0–1, reflecting the singularity of the fluctuation (Marshak et al.
1997). The larger the C1, the more intermittency in nonstationary data, and the more singular the fluctuations. According to Eq. (5) and Eq. (7), it can be seen that the premise of Eq. (9) here is that K(1)=0 (Lu, 2008).

### 3.3 IGWs and turbulence parameter

Based on the data during the rising stage, we use hodograph analysis to extract IGW parameters (Bai et al., 2016; Huang et al., 2018), with a height interval of 18–25 km, thereby obtaining parameters including vertical wavelength, horizontal
wavelength, intrinsic frequency, propagation direction (anticlockwise from y axis), kinetic energy, potential energy, and

momentum flux. In order to eliminate the error caused by the random movement of the balloon, the data is uniformly interpolated to an interval of 50 m. The total energy is the sum of kinetic energy and potential energy.

Based on Thorpe analysis (Ko & Chun, 2022; Thorpe, 1977; Wilson et al., 2011), the atmospheric turbulent layer is identified from the sorted potential temperature profile, thereby obtaining turbulence parameters including Thorpe length, turbulent layer thickness, turbulent kinetic energy dissipation rate, and turbulent diffusion coefficient. Optimal smoothing and statistical tests are used to distinguish between "overturn" caused by real turbulent motion and artificial "inversion" caused by instrument noise and balloon motion (Wilson et al., 2011). Since turbulence is highly intermittent, the turbulence parameters obtained here are derived from the regional average of non-zero values (turbulence exists) within the height range of 15–25 km of each profile.

## 4 Results and Discussion

### 4.1 Determination of scale interval

When no turbulence occurs (there is no $r$ slope at the tail of the third-order structure function), the calculated H1 and C1 both comes from the fitting interval of the GW subrange. When turbulence occurs (there is an $r$ slope at the tail of the third-order structural function), the fitting interval of turbulence and GWs should be distinguished, and the slope at the corresponding scale should be calculated separately. Taking into account the different separation distances of different data, the scale range corresponding to the calculated parameters will vary. However, in order to facilitate comparison, we use the separation distance $r$ closest to 500 m ($< 500$ m) as the turbulent outer scale $R_t$, and the separation distance closest to 6 km ($< 6$ km) as the gravity wave outer scale $R_w$, aiming to identify small-scale, high-frequency GWs with a spatial scale of several kilometers. The fitting intervals of turbulence and gravity waves are $[\eta, R_t]$ and $[R_t, R_w]$, respectively.

During statistical analysis, in order to compare the GWs that did not accompany the turbulence with the GWs that accompanied the turbulence, the calculated H1 and C1 are unified into the same fitting interval $[\eta, R_w]$. When turbulence occurs in the tail, the C1 value obtained from $[\eta, R_w]$ interval will also be larger, which means that C1 calculated over a wider range can also recognize the occurrence of turbulence. In order to obtain C1 in $[\eta, R_w]$ fitting interval from Eq. (9), it is necessary to ensure that $K(1) = 0$ (or approximately close to 0), thereby discarded unsatisfactory cases. Here $K(1)$ approximately close to 0 is defined as $K(1) < 0.01$. When $K(1)$ exceeds this value, it can be intuitively seen from the $K(q)$ curve that $K(1)$ and 0 have a certain distance. The physical explanation behind it is that the flat-floating trajectory is too irregular, or the actual detected wind speed has too many wild values (abnormalities from the positioning data).

The velocity increments $\delta u_L(r)$ is the key process for calculating the disturbance parameters from flat-floating data, and has shown good robustness within the separation distance of small-scale gravity waves (Figure A2). In fact, choosing the scale closest to 6 km (less than 6 km) can not only satisfy the statistical quantity of parameter results, but also ensure the robust of velocity increments on this scale. With the increase of the separation distance, the fluctuation of velocity increments becomes

more and more distinguishable. That is, too long a scale will cause significant differences in the velocity increments du at different data points, and the result will be no longer robust and cannot be used to calculate H1 and C1. Therefore, the selected SGW scale of 6 km will not be affected by the fluctuation of flat-floating height, as well as the swing of the balloon.

## 4.2 Quantification of atmospheric disturbance information

Taking the data from the Yichang site as an example, we illustrate how to identify the disturbance information from the flat-floating data. The multi-order structure function $S_q(r)$ is shown in Figure 3a. Using the $S_q(r)$ curve of $q=1$ for linear fitting, H1 can be obtained, with a value of 0.68. From the third-order structure function, a downscale energy cascade (from large to small scales) can be seen, with a $r^3$ slope indicating that no turbulence has been observed within the resolved resolution. Figure 3c is the relationship between the $q$-order singularity measure $\langle\varepsilon(r;x)^q\rangle$ and the separation distance $r$ in log-log coordinate. The curves $q=1$, $q=2$, $q=3$, $q=4$, and $q=5$ are given, from which the slope values can be calculated within the selected SGW scale (left of the black dashed line) as $-K(1)$, $-K(2)$, $-K(3)$, $-K(4)$, and $-K(5)$, respectively. Then the variation curve of K(q) with $q$ can be obtained in Figure 3d, where $q=0, 0.25, 0.5, …, 5$. The fitting slope of the $K(q)$ curve at $q=1$ is calculated from the $K(1)$ values corresponding to $q=0.75$, $q=1$, and $q=1.25$, and this slope vale is defined as intermittent parameter C1. Using the criterion proposed in section 3 for the identification of GW state, this case can be identified as a stable GW, and the GW scale quantified by (H1, C1) is 5.1 km.

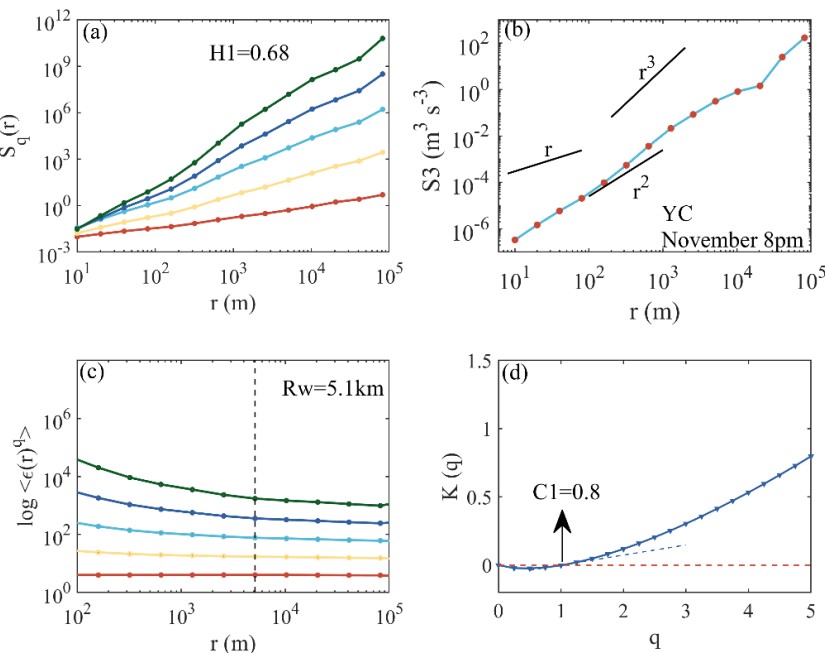

**Figure 3. (a) Multi-order structure function, (b) third-order structure function (the red dots represent negative values), (c) multi-order singular measure, (d) slope K(q) obtained from Yichang site on November 8. In Figure 3d, the red dashed line is K(q)=0, and the blue dashed line is the fit slope at K(1)**

Figure 4 shows cases for the coexistence of GW and turbulence and the unstable GW. The case for Yichang data at October 15 pm can be identified as a GW coexisting with turbulence, with a scale of 5.1 km. The GW is quantified as (0.59, 0.10), where the first value is H1 and the second value is C1. The case for Yichang data at November 10 pm can be identified as an unstable GW, and the GW is quantified as (0.50, 0.12), with a scale of 3.1 km.

By comparing the case results of Figure 3 and Figure 4, multi-order structure function (third-order structure function) can be found to have the spectral shape differences on certain scales, which mainly comes from the intervals with significant inclinations accompanied by a relatively large increase or decrease in the speed increment $u_L(r)$ on these intervals (Figure A3). Since $S_q(r) = \langle |\delta u_L(r)|^q \rangle$, when the curve of $S_q(r)$ at a certain separation distance r has an obvious inflection point, it means that there is a sudden increase or decrease of some velocity increment in the set of all velocity increment at this scale (He et al., 2022).

For the stable gravity wave (Yichang site on November 8), the flat-floating trajectory moves approximately along a quasi-straight line (Figure A3b), reflecting a relatively single physical flow region, which indicates that the internal instability of atmospheric wind field fluctuations is relatively weak. For the coexistence of gravity waves and turbulence (Yichang site at October 15 pm) and the unstable gravity wave (Yichang site at November 3 am), the flat-floating trajectory has been significantly deflected (Figure A3d and A3f), indicating that the detection area contains different physical flow regions, which means that the internal instability of atmospheric wind field fluctuations is relatively strong. Obviously, this also caused the sawtooth structure in the spectral shape and the inconsistency in the energy cascade direction of the third-order structure function.

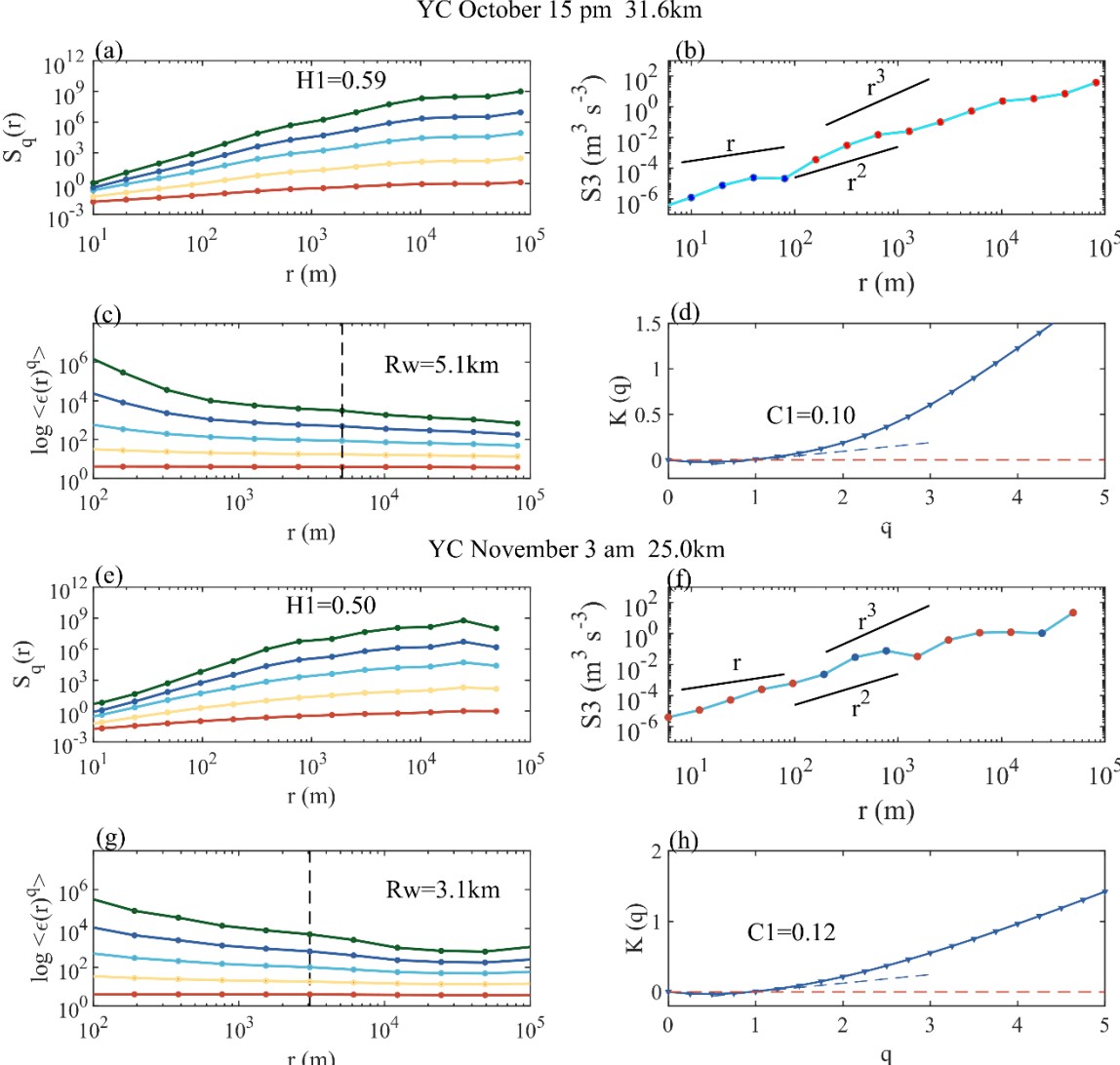

**Figure 4. (a) Multi-order structure function, (b) third-order structure function, (c) multi-order singular measure, and (d) slope K(q) obtained from Yichang site at October 15 pm, (e) multi-order structure function, (f) third-order structure function, (g) multi-order singular measure, and (h) slope K(q) obtained from Yichang site at November 3 am**

Therefore, when the stratospheric disturbance information is relatively abstract, the disturbance intensity can be quantified using (H1, C1) as a reference for mutual comparison. Considering that the calculation of wind speed comes from the coordinates of the positioning system, it is necessary to make sure that there is no wild value interfering with the results. The difference of positioning coordinates in adjacent time can identify the abnormal situation of positioning data, that is, whether there are obvious wild values in the difference of longitude or latitude. Figure A4 shows the cases for abnormal and normal positioning data, and these abnormal cases are screened out. Figure 5a and 5b show the histogram of Hurst parameters and

intermittent parameters of all data from the six sites, respectively. In summer, the H1 (C1) value is mainly concentrated in the range of 0.6–0.8 (0.08–0.16), while in autumn, the H1 (C1) value is mainly concentrated in the range of 0.5–0.7 (0.06–0.14). Compared with summer, stratospheric wave disturbances in autumn have a lower H1 and C1 distribution. It is reasonable to have a lower H1 distribution in autumn, since the flat-floating trajectories of the six sites in autumn are more irregular. The obvious change in the trajectory (away from the previous straight direction) indicates that the detected data contains different

physical flow regions, suggesting the internal instability and multifractal characterizations of the background wind field fluctuations (Lu and Koch, 2008).

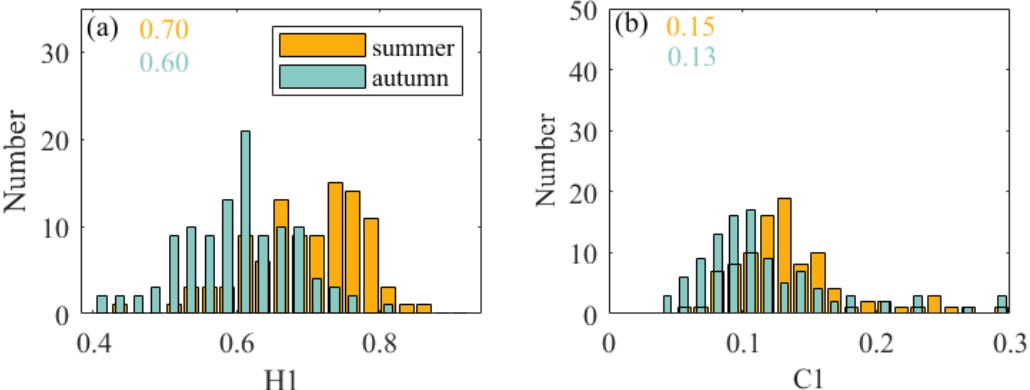

**Figure 5. Histogram of (a) Hurst index and (b) intermittent parameter from all flat-floating data over the six sites4.3 Statistical results of disturbance parameters**

The distribution of inertial gravity wave and turbulence parameters is shown in Figure 6. The wavelength, intrinsic frequency, and energy of IGWs in summer and autumn show no obvious differences. The momentum flux in summer has a significant positive shift, the net zonal momentum flux is eastward with easterly winds dominated in the stratosphere. The dominant propagation directions of IGWs in summer and autumn are northeast and southwest respectively, due to the effect of "critical layer filtering" (Eckermann, 1995). The background wind field filters out gravity waves propagating in the same

direction, and passes through gravity waves propagating in the opposite direction. For disturbances from small-scale turbulence, there is no obvious difference between the Thorpe length and turbulence thickness in summer and autumn. In autumn, the turbulent kinetic energy dissipation rate and turbulent diffusion coefficient have a more ideal Gaussian distribution with smaller peak value, indicating that the wave source is more single and the turbulence activity is weaker than that in summer. The deviation of turbulence peaks in different studies may come from the intermittency of turbulence, sensor performance, and

regional source characteristics (Ko and Chun, 2022; Zhang et al., 2019; Lv et al., 2021).

   In this paper, the vertical wavelength of the IGW is concentrated in the range of 1–3 km, which is close to the scale of the stratospheric IGW in China (1.5–3 km) observed by radiosonde data (Bai et al., 2016). In our results, kinetic energy and potential energy of IGW are concentrated at 2–6 J/kg and 0–2 J/kg, respectively. In the tropics, by contrast, the kinetic energy of stratospheric IGW has already exceeded 10 J/kg (Nath et al., 2009), indicating more intense wave activity at lower latitudes.

The turbulent kinetic energy dissipation rate $log_{10}\varepsilon$ is between -5 and -2 from RTISS, which is comparable to those obtained based on radiosonde data in the United States from $-4$ to $-0.5$ $m^2 s^{-3}$ (Ko and Chun, 2022) and in Guam from $-6$ to $0$ $m^2 s^{-3}$ (He et al., 2020a).

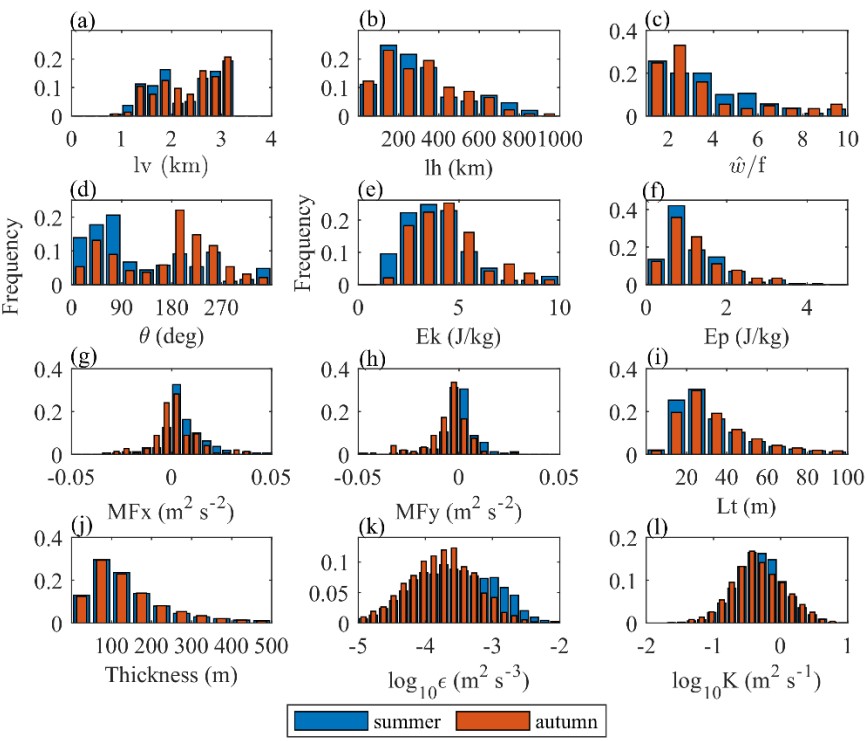

**Figure 6. Histogram of disturbance parameters for IGWs including (a) vertical wavelength, (b) horizontal wavelength, (c) intrinsic**
**frequency, (d) horizontal propagation direction, (e) kinetic energy, (f) potential energy, (g) zonal momentum flux, and (h) meridional momentum flux. Histogram of disturbance parameters for turbulence including (i) Thorpe length, (j) Turbulent layer thickness, (k) Turbulent kinetic energy dissipation rate, and (l) turbulent diffusion coefficient**

The results of H1 and C1 over the six sites are shown in Figure 7. Compared with the coexistence of GW and turbulence or unstable GW, stable GW tends to have a larger H1 and a smaller C1. The cases in red rectangles are the detection of adjacent
times when the flat-floating height is close, which is convenient to compare the third-order structure functions and the wind speed disturbance behind the different (H1, C1), the result is shown in Figure 8. The value of H1 is related to the smoothness of the data series, that is, the denser the wave packets superimposed on the fluctuation trend, the smaller the H1. The value of C1 is related to the singularity degree of the data series, that is, the more disturbances that deviate significantly from the mean state in a local region, the larger the C1 value. The protruding part of the purple circle in Figure 8 is the local area of the
disturbance sequence (the one with the larger C1 value) that causes the intermittent parameter to be too large. Taking two cases of GZW as examples (Figure 8), compared with the detection at October 17 pm, the detection at October 20 pm has smaller H1 and larger C1. The data series at October 20 pm is rougher with denser wave packets, and there are more obvious strong perturbations deviate from the mean state in the local area. This is the first time that a relatively comprehensive (multi-site,

multi-time) result of stratospheric atmospheric disturbance information in the horizontal direction has been given by balloon observation in China, which can provide an intuitive reference for the cognition of the stratospheric atmospheric environment.

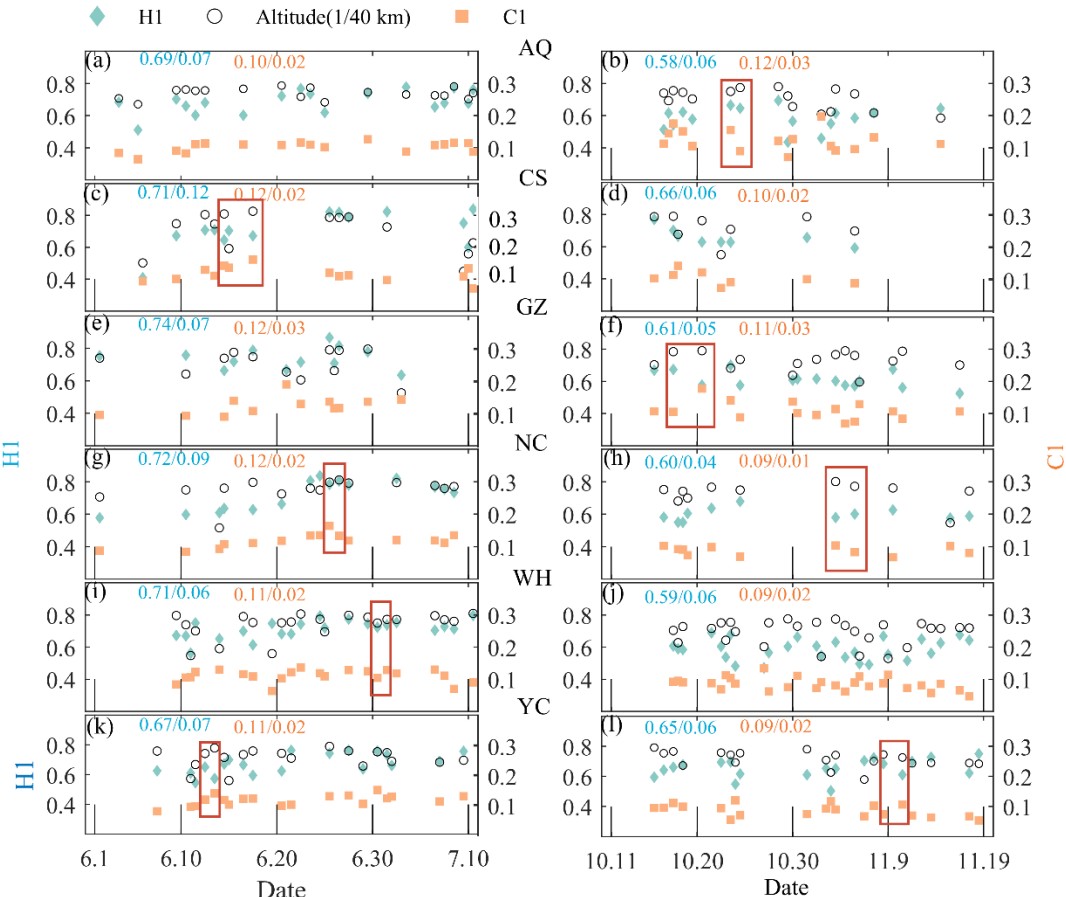

**Figure 7. The atmospheric disturbance parameters (H1, C1) and the corresponding average flat-floating height (scaled to 1/40) obtained over the six sites in summer (left panel) and autumn (right panel), the mean and standard deviation of H1 and C1 are marked in blue and yellow, respectively**

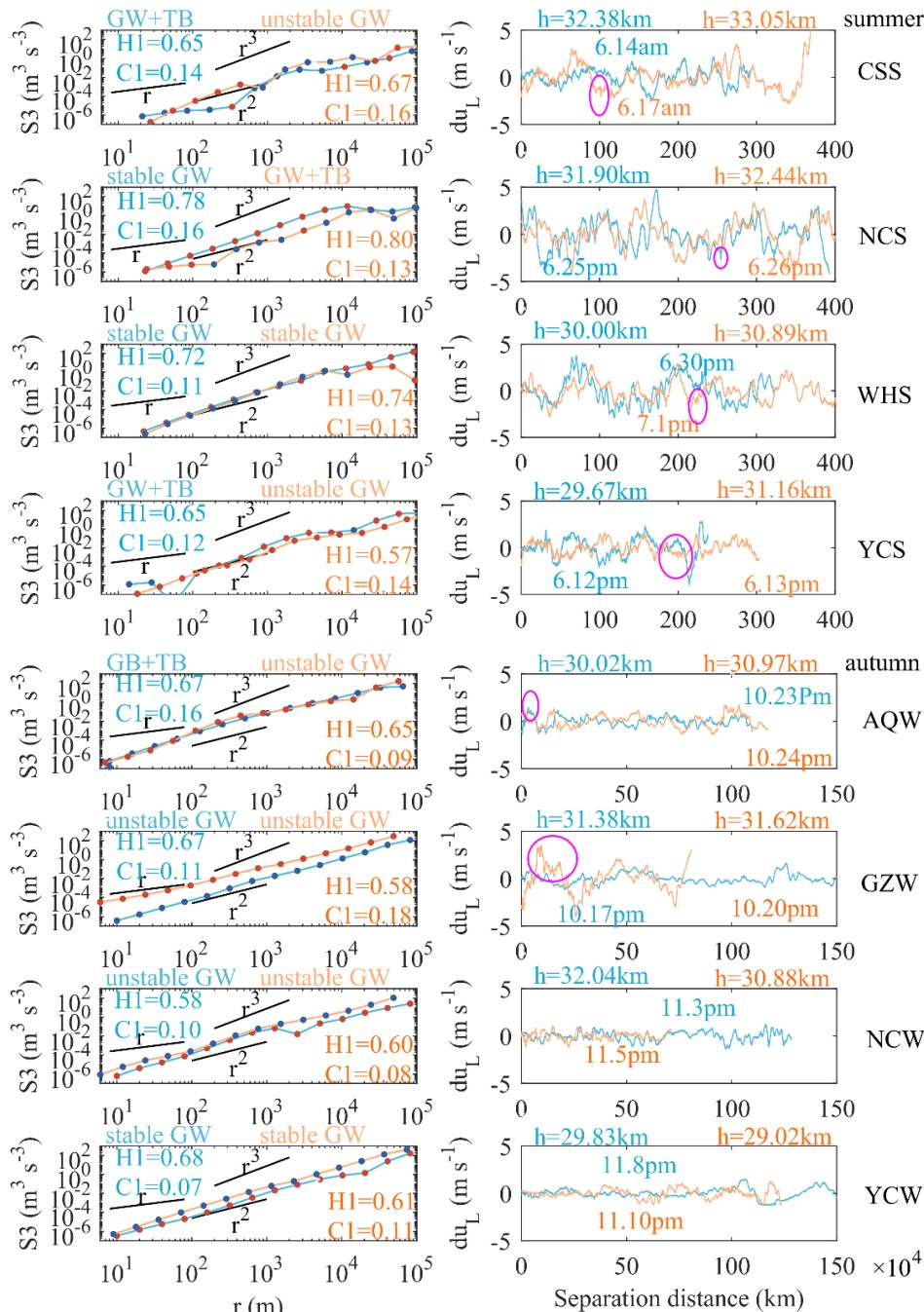

**Figure 8.** The third-order structure function (left panel) and the longitudinal velocity component perturbation (right panel) for the selected cases in the corresponding red rectangles. In order to better compare the roughness and singularity of the velocity component, the longitudinal velocity component perturbation is used here after removing the background field by using the fourth-order polynomial fitting. The protruding part of the purple circle is the local area of the disturbance sequence (the one with the larger C1 value) that causes the intermittent parameter to be too large



## 4.3 Potential links between multiscale fluctuations

Although there are different methods for quantifying wave disturbances, linking detection results from different profiles (for example, in the vertical and horizontal directions) is still a challenge and an observation gap. Taking the detection results from RTISS as an opportunity, the possible connection between wave disturbances obtained by different quantitative methods is discussed, and the result is shown in Figure 9. It should be noted here that the wave disturbance extracted from the flat-floating data are small-scale, high-frequency GWs with a spatial scale of several kilometers, while the wave disturbance extracted from the rising data are IGWs with a spatial range of several hundred kilometers. There is no clear linear correlation between H1 and C1 (Figure 9a). C1 can reflect the intensity of turbulence mixing and is highly intermittent and random, which is not related to height (Figure 9b). In contrast, there is a significant positive linear correlation between H1 and height (Figure 9c). As height increases, the entire data series tends to be smoother.

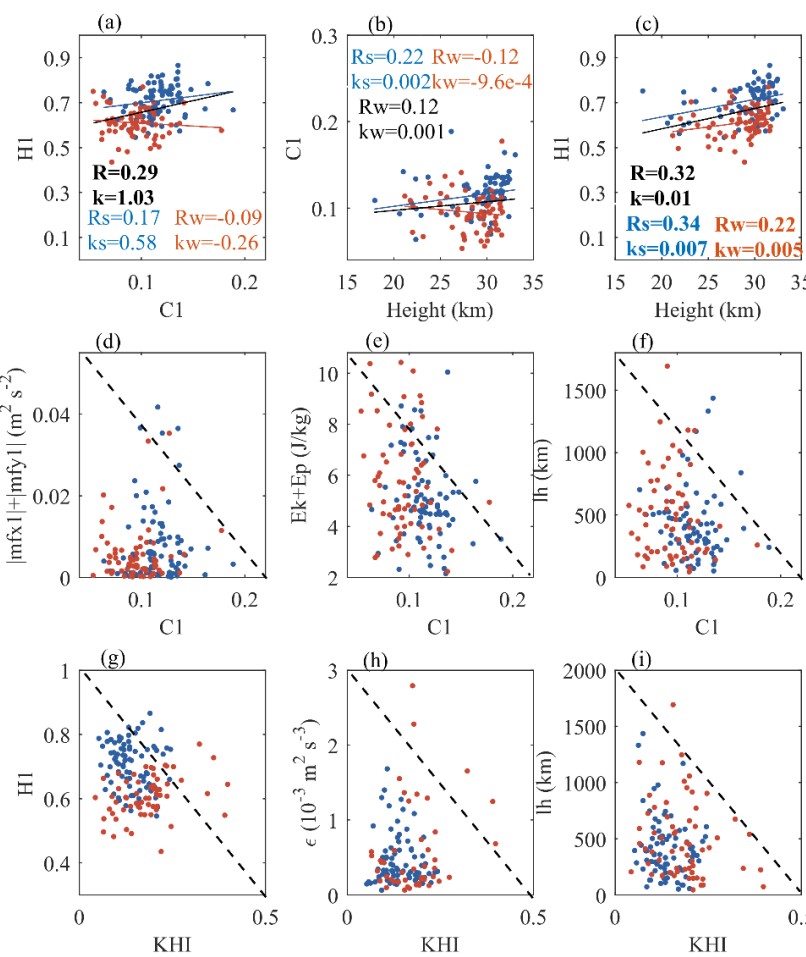

**Figure 9.** Scatter plots of (a) H1 versus C1, (b) C1 versus height, (c) H1 versus height, (d) momentum flux versus C1, (e) total energy versus C1, (f) horizontal wavelength versus C1, (g) H1 versus KHI (ratio of $0 < Ri < 0.25$), (h) $\varepsilon$ versus KHI, and (i) horizontal wavelength versus KHI. Blue and red dots represent summer and autumn. The blue, red, and black lines in (a)–(c) represent linear fitting results of summer, autumn, and all data, respectively

Due to the limitations of the sample size and the different detection objects, the linear correlation between these variables from Figure 9d–f may not be statistically significant, so we pay more attention to the change trend between them. With the increase of C1, the momentum flux, total energy, and horizontal wavelength of IGWs are more concentrated in a lower range (Figure 9d–f). Next, we consider that the wave disturbance in the stratosphere is likely to be related to the Kelvin Helmholtz instability (He et al., 2020b; Lu and Koch, 2008),. The ratio of $0 < Ri < 0.25$ between 15 and 25 km represents the instability. As KHI increases, the horizontal wavelength of IGWs decrease (Figure 9i), while the data sequence of SGWs tend to be rougher (Figure 9g). Although the quantity of large C1 values (>0.15) is relatively rare (the detected disturbances with strong intermittence are still small probability events in the entire sample), it is still possible to see that the enhanced C1 is accompanied by the weakened momentum flux, energy, and horizontal wavelength of IGWs.

From the above results, it can be seen that the increased instability of SGWs in the stratosphere will be accompanied by the weakening of IGWs below. The KHI that appears in an unstable shear due in part to IGWs (Abdilghanie and Diamessis, 2013) is likely to be the excitation source of small-scale, high-frequency GWs propagating to higher altitudes. This phenomenon has also been confirmed by numerical simulations in the mesosphere and higher altitudes (Dong et al., 2023).

## 4.4 Relation between parameter space and ozone transport

The transport of ozone and its changing trends is one of the important issues concerned in stratospheric research, which is closely related to the atmospheric radiation balance and global warming (Tian et al., 2023; Fei Xie et al., 2016; Jiankai Zhang et al., 2022; He et al., 2023). The ozone and potential vorticity (PV) have good consistency, which can be regarded as good indicators for studying the stratospheric material transport process (Allaart et al., 1993; Newell et al., 1997). Considering that the GW process plays an important role in the transport of ozone between the upper and lower layers (Gabriel, 2022), we aim to explore whether there is a direct connection between the quantitative indicator of wave disturbance and ozone.

Based on the ERA5 reanalysis data, the ozone mass mixing ratio (OMR) and PV at different pressure layers that matched the detection are selected. Specifically, the ERA5 data at 00UTC and 12UTC within the longitude and latitude range of the selected flat-floating stage are screened, and the value after regional average is used as the reanalysis data result corresponding to the flat-floating detection at that time. The matching results of different air pressure layers (150 hPa, 125hPa, 100 hPa, 70 hPa, 50 hPa, 30 hPa, 20 hPa, 10 hPa, 5 hPa, 3 hPa) are calculated.

Figure 10 shows the possible connection between C1 and these two indicators (OMR and PV). The pressure layers selected here correspond to the height above (10 hPa) and below (100 hPa) the flat-floating interval (20–30 km), in order to distinguish them from the height range where small-scale GWs are detected in the lower stratosphere (100 hPa), there is a significant positive correlation between ozone and PV, while in the middle stratosphere (10 hPa), there is a significant negative correlation between the two. For SGWs detected during flat-floating stage, the larger the C1, the weaker the PV in the stratosphere, accompanied by the reduction of IGWs (Kalashnik and Chkhetiani, 2017). This is consistent with the result that the higher C1 corresponds to the lower IGW energy below in Figure 9. The more intermittency of SGWs, the less (more) ozone below (above), thereby forming an enhanced ozone transport between them. In the process of area averaging, there are usually

only two or three ERA5 data points within the latitude (longitude) range of the flat-floating trajectory. However, there are still

some cases without matched ERA5 data, we extend the latitude (longitude) range to a width extending 0.25° north (east) and

south (west) from the center point of the trajectory. In this way, ERA5 data and trajectory can be matched as much as possible

under the premise that there is data in the matching area.

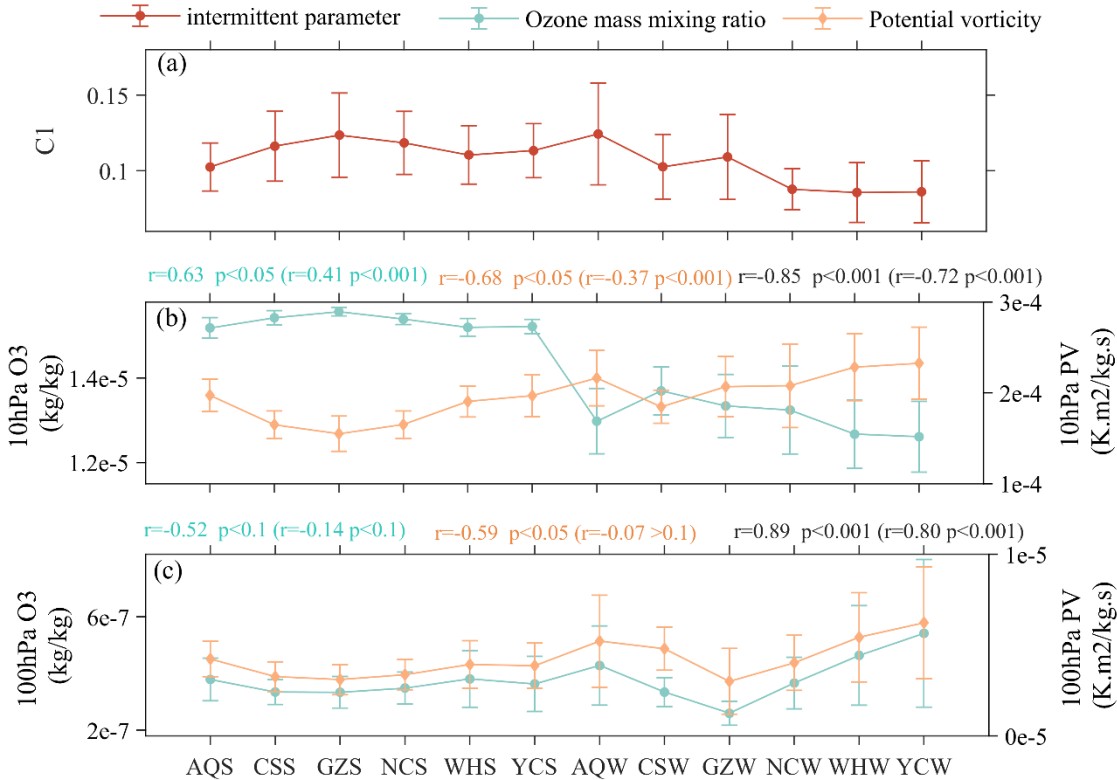

**Figure 10. The error bar diagram of (a) intermittent parameters C1, the ozone mass mixing ratio (OMR), and potential vorticity**
**(PV) at (b) 10hPa, (c) 100hPa pressure layers in summer (S) and autumn (W), showing a total of 12 clusters over the six sites. The**
**blue, yellow, and black annotations marked at the top of the subgraph indicate the Pearson correlation coefficient and significance**
**level for OMR versus C1, PV versus C1, and OMR versus PV, respectively. Outside the brackets is the correlation of the average**
**values of the 12 clusters (12 values), inside the brackets is the correlation of all cases of the twelve clusters**

The mechanism diagram of ozone transport and energy transfer is shown in Figure 11. The significant positive (negative)

correlation between C1 and ozone concentration in the lower (middle) stratosphere further support the argument that SGW

may affect the vertical transport of ozone (right part of Figure 11). The stratospheric SGWs detected here are closely related

to KHI, and previous studies have also confirmed this (He et al., 2020b; Lu and Koch, 2008). The transport capacity of IGWs

on ozone is weakened due to the critical layer filtering during its upward propagation. In contrast, the high-frequency SGWs

can propagate to higher altitudes (Dong et al., 2023). Ozone transport is closely related to the SGWs between 100 hPa and 10

hPa, corresponding to the weakening of IGWs in the lower stratosphere (100 hPa) and the enhancement of SGWs excited by

KHI. SGWs with higher phase velocities would propagate upward without encountering critical level and thus complete the

ozone transport to the middle stratosphere (10 hPa) (Heale and Snively, 2015; Li et al., 2020; He et al., 2022b). The enhanced intermittency is accompanied by the weakening of IGW energy below, which also reveals the possible energy transfer from large-scale to small-scale waves.


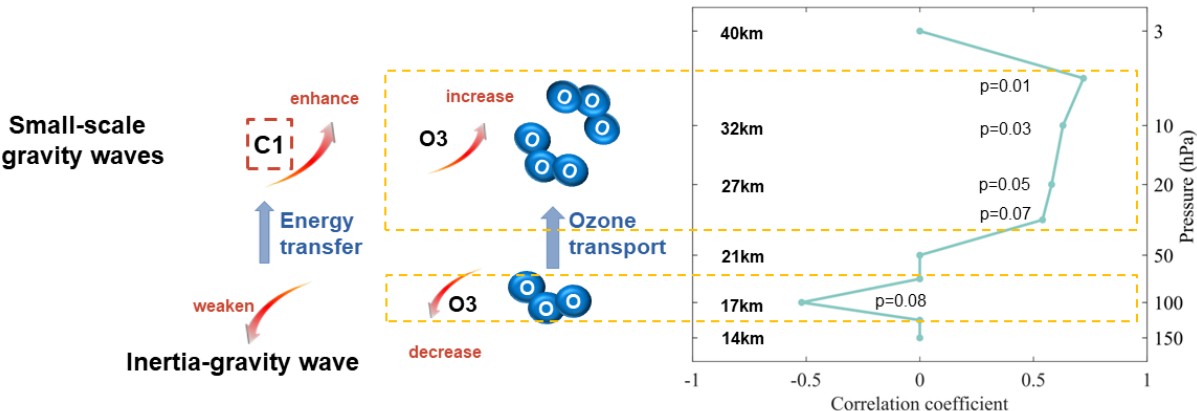

**Mechanism Diagram of Ozone Transport and energy transfer**

**Figure 11. The mechanism diagram of ozone transport and energy transfer. Right part shows the vertical distribution of correlation coefficient between the OMR and C1 in summer and autumn (a total of twelve clusters) over the six sites at different pressure layers. When the correlation for OMR versus C1 of the average values of the 12 clusters (12 values) and of all cases are both statistically significant (p < 0.1), it is considered that the small-scale GW disturbance is closely related to the change in ozone concentration on the corresponding pressure layers, otherwise the correlation coefficient is set to 0. For the pressure layers with significant correlation coefficient, the significance level p value corresponding to the 12 clusters is marked in the figure**

### 4.5 Calculation for a single physical flow regime

Two scales which shown as the inconsistency in the energy cascade direction are related to different physical flow regimes (Lu and Koch, 2008). In balloon observations, this different physical flow regimes will be represented by curved (non-linear) trajectories. Therefore, in order to retain this recognition of different physical flow regions, zonal or meridional projection is selected (which can decompose the curved trajectory into zonal or meridional), as the results shown above. In this section, we also use the method of linear fitting to show the calculation results of a single physical flow regime.

The YC case on October 15th is taken as an example to illustrate this method, shown as Figure 12. In order to ensure quasi-linear fitting, the region that can be approximated as a straight line for linear fitting is selected from the original flat-floating trajectory. The selected period is represented by the red rectangular box in Figure 12a. Then the data part that can be processed by line fitting is obtained in Figure 12b. By decomposing the zonal and meridional wind components into a new coordinate system (the X-axis is parallel to the fitted line), the longitudinal (along the separation distance direction) and transverse (normal to the separation distance direction) velocity components can be obtained (Figure 12c-d).

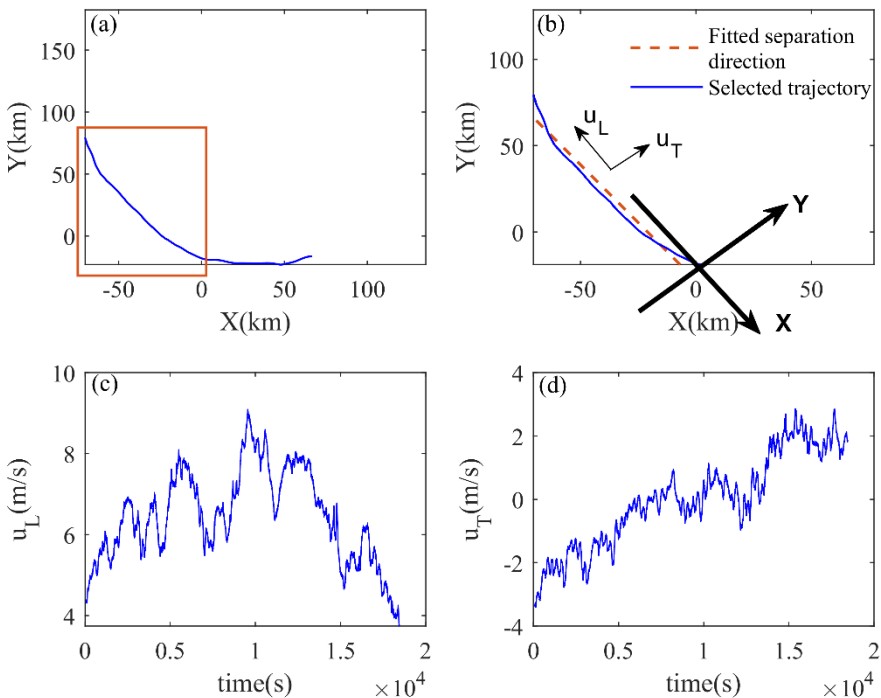

**Figure 12. The trajectory in the XOY plane (a) before and (b) after quasi-linear fitting, (c) the longitudinal (along the fitted line) velocity $u_L$ and (d) the transverse (normal to the fitted line) velocity $u_T$ after quasi-linear fitting from Yichang site at October 15 pm**

Furthermore, the third-order structure function and slope K(q) curve in the single physical flow region are obtained, as
shown in Figure 13. Compared with the zonal projection of the multi-physical flow regime (Figure 4b and 4d), the calculated
results of the single physical flow regime may be different on both H1 and C1, especially H1. The reason for this is that in the
process of linear fitting, partial trajectories that deviate significantly from the straight line are omitted. According to Eq. (1) in
the manuscript, the inconsistency between the convergence and divergence of velocity on adjacent scales leads to internal
instability. The balloon itself moves with the wind, so when there is a sudden change in the velocity field, the flat-floating
trajectory will naturally change. If the trajectory direction is relatively single (single-physical flow region), the linear fitting
method can actually get more accurate results. However, the problem is that because many curved trajectories are rounded out
after screening, the results obtained are not suitable for internal comparison. And irregularly curved trajectories may also
contain important disturbance information. Compared with the best linear fitting of the single-physical flow region, the zonal
or meridional projection in the multi-physical flow zone can be said to be a compromise method. Not only can more samples
be retained, but also the disturbance information behind the curved/irregular trajectories can be retained.

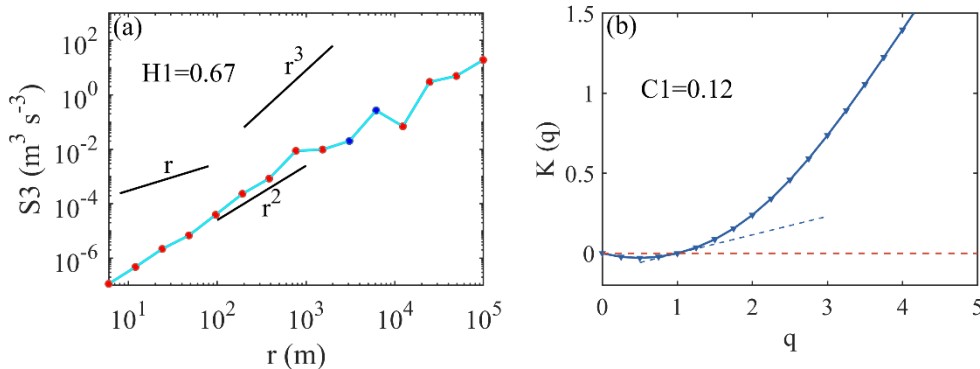

**Figure 13. (a) third-order structure function, and (b) slope K(q) obtained from Yichang site at October 15 pm5 Summary and Conclusions**

## 5 Summary and conclusion

Based on the round-trip intelligent sounding system (RTISS) released in China, we conducted a systematic analysis on the atmospheric disturbance information in the stratosphere. Using the structure function and singular measurement, the parameter space (H1, C1) is calculated to describe the nonstationarity and intermittency of atmospheric dynamic processes. The physical process corresponding to the stratospheric SGWs is mapped to this parameter space, realizing the comparison of disturbance characteristics between different cases (different in flat-floating height, time and area). There is a significant linear

relationship between H1 and height. As the height increases, the nonstationarity (roughness) decreases. In contrast, the distribution of C1 is more random and independent of height, and the intensity of turbulence mixing and SGWs at different altitudes can be compared.

The continuous detection from rising and flat-floating stages realizes the seamless capture of stratospheric SGWs and IGWs below them. By analyzing the correlation between the parameters calculated by multiple-scale disturbances, the

connection between IGWs and SGWs is qualitatively revealed. The results show that the enhancement of SGWs is accompanied by the weakening of IGWs activity below, and the generation of these SGWs is related to KHI. In addition, we explored the role of GWs in stratospheric ozone transport based on the potential relationship between intermittent parameter C1, potential vorticity and ozone, and found that the enhancement of SGWs is conducive to the transport of ozone from lower stratosphere to higher altitudes, although the length of this path is limited due to the wave dissipation. This is the first time that

high-frequency, long-duration in situ detection method has been used to discuss the role of stratospheric multi-scale disturbances in energy transfer and material transport in China. The introduction of flat-floating information provides a new idea for the study of stratospheric dynamic processes, while the three-stage detection supplements the research of stratospheric-tropospheric interaction (Scaife et al., 2012; Niu et al., 2023).

Encouragingly, the quantitative description of SGWs in the stratosphere using (H1, C1) has shown a possible connection
with larger-scale IGWs and smaller-scale turbulence, and a potential relationship between it and stratospheric ozone transport
can be found. Of course, given the limited number of samples and the different perturbation extraction methods in vertical and
horizontal directions, the potential connection between these multi-scale fluctuations may not be significant. However, the
linear relationship between disturbance from IGW and SGW can be significant if only kinetic energy is considered (the
calculation of the disturbance parameter in SGW is derived only from the wind speed), as shown in Figure A5. This also
indicates that the enhancement of SGW is indeed accompanied by the weakening of IGW. Besides, regardless of whether the
flat-floating trajectory has been linearly fitted or not, this significant linear relationship exists.

The SGWs captured by the flat-floating balloon discussed are mainly concentrated in the stratospheric altitude range of
20 km–30 km. However, it should be noted that this does not mean that the SGW activity outside this altitude range can be
ignored (including the upward-propagating of SGW inside the altitude range and the undetected SGWs outside the altitude
range), which is a possible reason for the significant positive correlation between C1 and ozone at higher altitudes (the positive
correlation on the 5 hPa pressure layer in Figure 11). Considering that an initial ascent of an air parcel can lead to an increase
(decrease) in ozone above (below) compared to the surrounding atmosphere, the general positive correlation between C1 and
ozone within the height range where small-scale GWs are detected shows that the propagation direction of SGW is mainly
upward.

The use of RTISS provides an opportunity for related research: that is, it is possible to achieve quasi-seamless detection
of the atmospheric structure from both the vertical and horizontal directions inside the stratosphere at the same time. The
relatively high resolution is also conducive to better capturing the fine structure of atmospheric disturbances. Taking the inertial
gravity waves and small-scale gravity waves studied in this manuscript as an example, the effective capture of different
disturbances in continuous time based on RTISS on different cross-sections is impossible to achieve with other single
measurement methods. Due to the limitation of the sample size and the differences in calculation methods, there may be some
not completely significant relationships in the discussion of different wave types and their relationship with ozone. The
exploration of stratospheric atmospheric disturbances and material transport using this new detection method is still worthy of
continuous follow-up and improvement. As valid samples gradually accumulate, these relationships may become more
significant and robust.

Our results reveal the important role of stratospheric SGWs in material transport and energy transfer, and demonstrate
the potential ability of physical parameter space (H1, C1) in stratospheric dynamics research. Follow-up research is worth
continuing, using the detection results of RTISS in more regions with longer periods, to improve the understanding about the
statistical characteristics and regional differences of stratospheric disturbance information. Besides, potential connections that
may exist between this parameter space and other atmospheric components (such as water vapor, carbon dioxide, methane,
etc) transported in the stratosphere also deserves further attention.

**Appendix A**

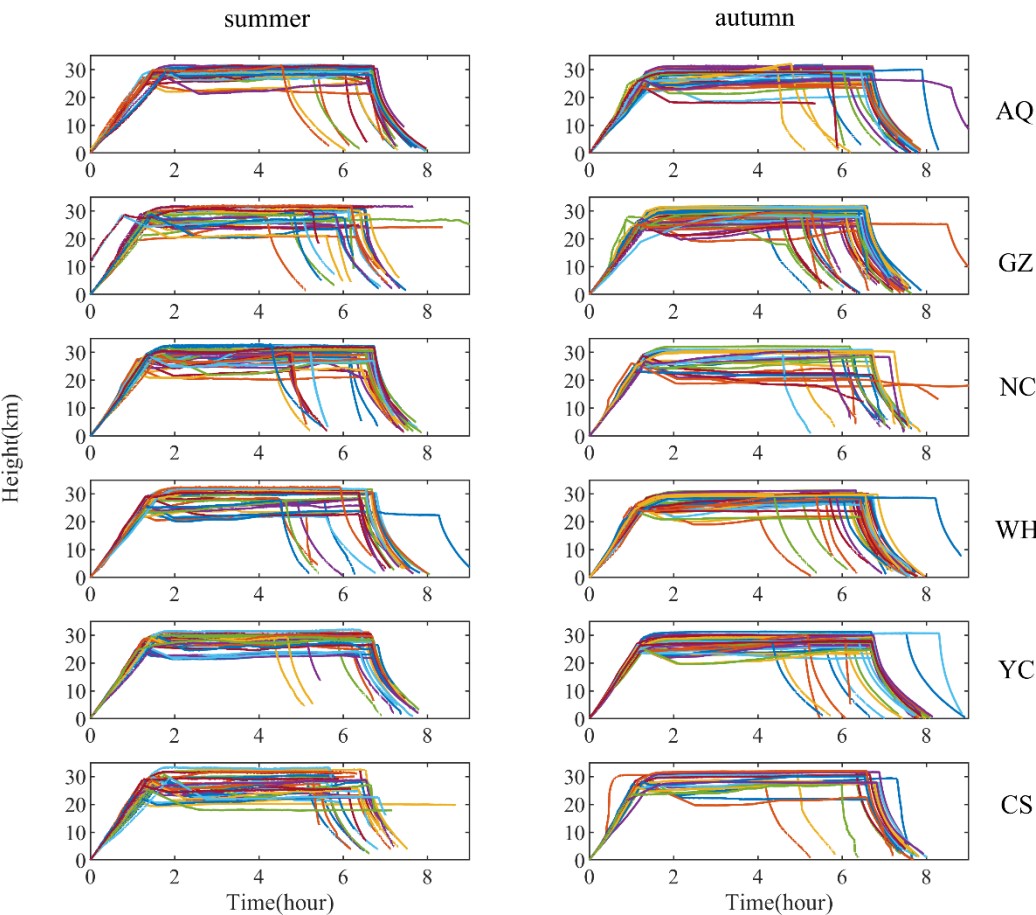

**Figure A1. Time-height curves in summer (left) and autumn (right) during the entire detection process for RITSST detections at six sites**


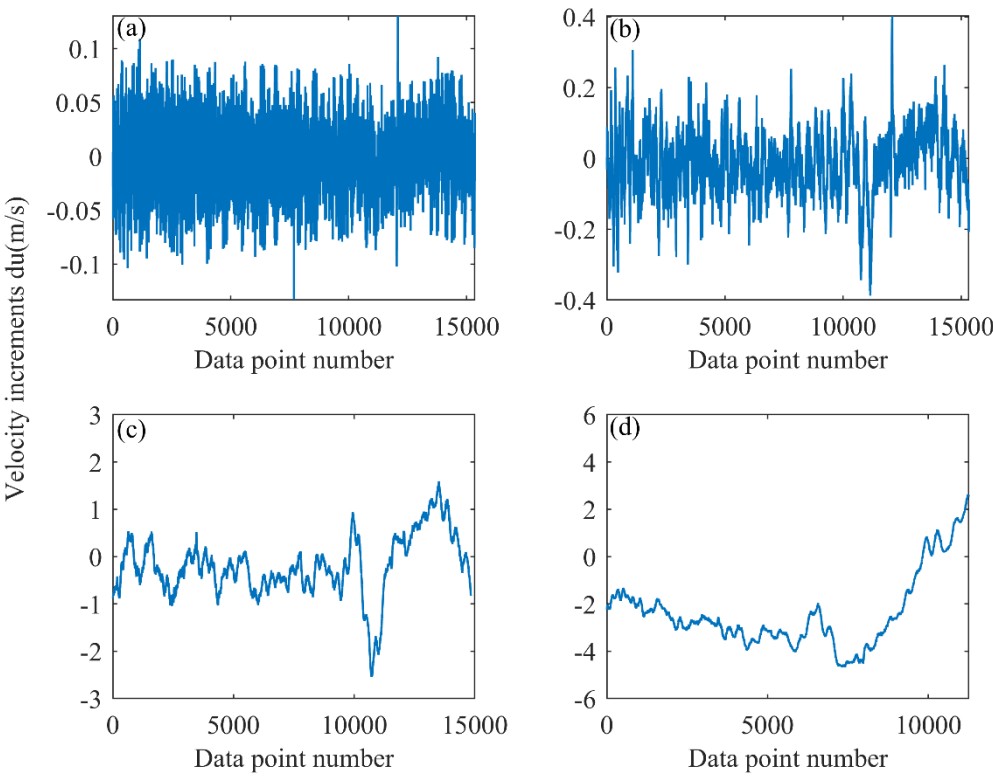

**Figure A2. Velocity increments calculated from beginning to end in the data series from Yichang site on November 8, where the separation distances are (a) 44 m, (b) 352 m, (c) 5600 m, and (d) 45 056 m, respectively**

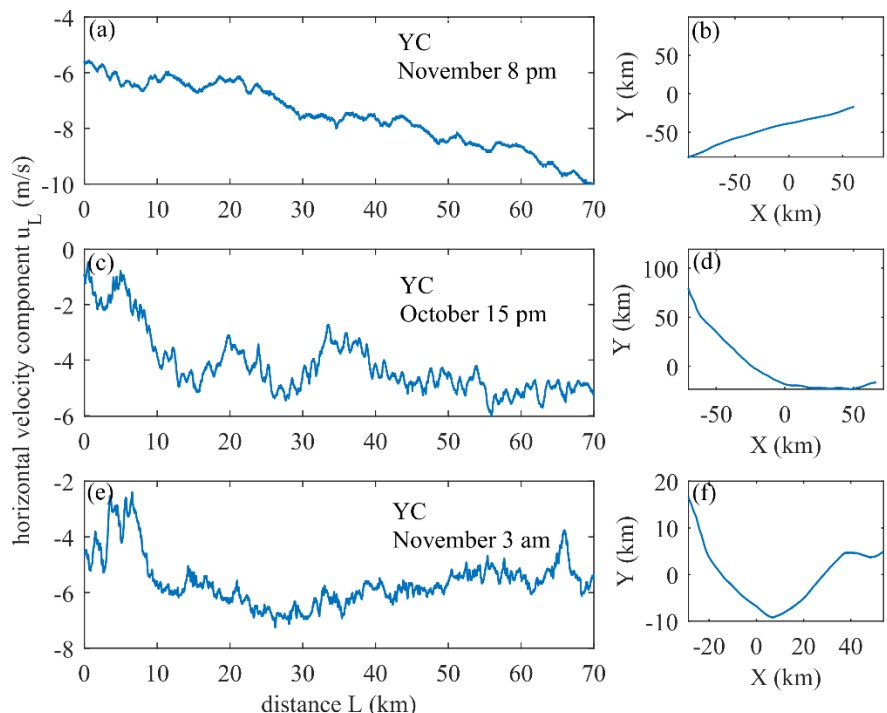

**Figure A3. The variation of horizontal velocity component uL along the zonal separation distance (left panel) and flat-floating trajectory (right panel) from three cases**

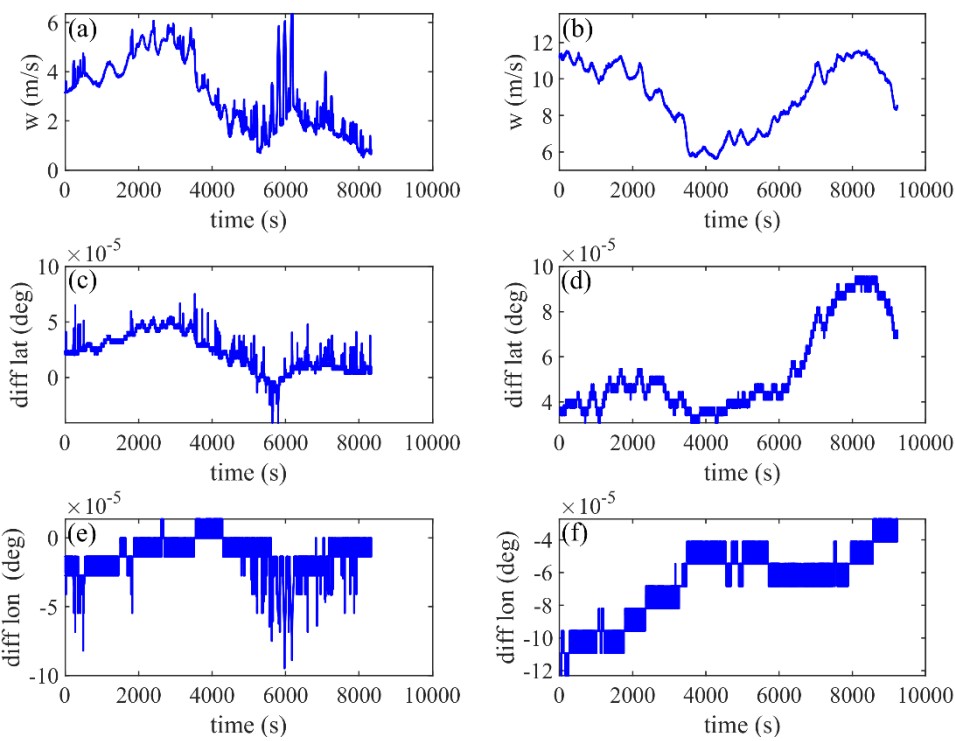

**Figure A4. (a) wind velocity, (c) latitude difference, and (e) longitude difference for the case where the positioning data is abnormal, and (b) wind velocity, (d) latitude difference, and (f) longitude difference for the case where the positioning data is normal**

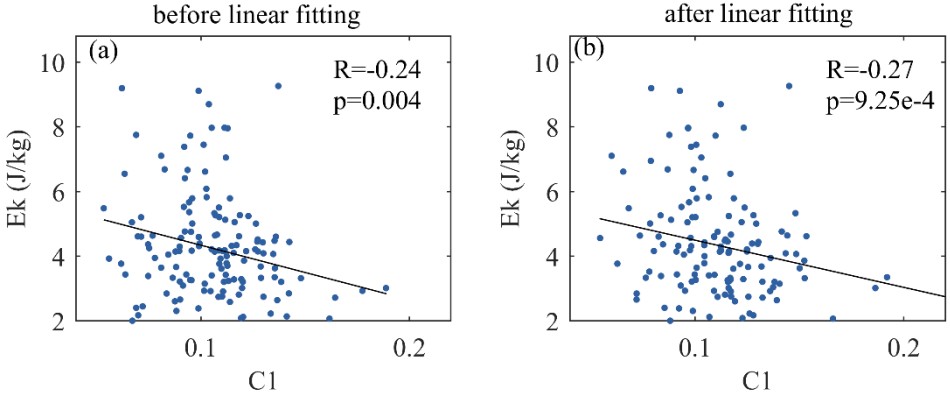


**Figure A5. Scatter plots of Ek versus C1 (a) before linear fitting and (b) after linear fitting**

**Code and data availability**

ERA5 dataset is publicly available through https://doi.org/10.24381/cds.adbb2d47. The procedures and the data files needed to recreate the figures can be download in 4TU.Centre for Research Data. Software and data used in this manuscript are deposited through https://doi.org/10.4121/7c37ae88-0215-4803-8403-57e48088ff0f.

**Author contributions**

Zheng Sheng, Yang He, Xiaoqian Zhu, and Mingyuan He initiated the study. Zheng Sheng designed the scheme, Yang He analyzed data and drew figures, Yang He and Zheng Sheng wrote the manuscript. All the authors interpreted results and revised the manuscript.

**Competing interests**

The contact author has declared that neither of the authors has any competing interests.

**Acknowledgments**

This work was supported by the National Natural Science Foundation of China (Grant 42275060), the Hunan Outstanding Youth Fund Project (Grant 2021JJ10048), and the Postgraduate Scientific Research Innovation Project of Hunan Province (Grant CX20220046). Thanks for the support provided by "Western Light" Cross-Team Project of the Chinese Academy of Sciences, Key Laboratory Cooperative Research Project. Additionally, helpful comments by the editors and the specific anonymous reviewers are gratefully acknowledged.

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
