# Peer review of "Identification of stratospheric disturbance information in China based on round-trip intelligent sounding system"

_EGUsphere, 2023_

## Author Comment (AC1)

The editorial support team
*Atmospheric Chemistry and Physics*
October 15th, 2023
Subject: Revision of manuscript egusphere-2023-1608

Dear Editors and Reviewers:

Thank you for your letter and for giving us the opportunity to revise our manuscript on "Identification of stratospheric disturbance information in China based on round-trip intelligent sounding system" [Paper # egusphere-2023-1608]. We have carefully reviewed the comments and have revised the manuscript accordingly. Our responses are given in a point-by-point manner below. Changes to the manuscript are shown in the revised manuscript with "track changes".

Sincerely,

Yang He

E-mail: heyang12357@sina.com

Corresponding author: Zheng Sheng
E-mail: 19994035@sina.com

**Response to Reviewer #1:**

General comments:

The paper "Identification of stratospheric disturbance information in China based on round-trip intelligent sounding system" by Yang He et al. presents statistical analysis of observational data obtained from specific balloon measurements at six sites in China. The extraction and analysis of the stratospheric gravity wave disturbance is an important topic, as more knowledge on gravity waves and their interactions is needed for the improvements of gravity wave parametrisations. Regarding the methodology of the paper, I am not entirely satisfied. During the procedure, for example, steps leading to reduction of the datasets were applied, without any discussion and verifications on possible impacts on the results. Also, I have the impression that too strong implications are sometimes deduced from rather ambiguous results. The structure of the paper might be improved by moving the methodology out from the result section and adding a discussion section. Finally, as for the language and notation, I think the paper needs to be carefully read through and cleaned.

**Response**: Thanks for appreciating our contribution and performing such an insightful and detailed review. We have made targeted revisions and replied in accordance with the specific opinions you give later. Your professional opinions are very valuable for improving the quality of our manuscript. We also look forward to your feedback on our improved work.

Specific comments:

1) L91: How large is the integer multiple of the swing period? Could smoothing have an effect on elimination of GWs?

**Response**: We select the appropriate smooth fitting interval to eliminate the pendulum motion. An integer multiple of the swing period is used as the smooth fitting interval, and

the symmetry of the swing is used to compensate for the swing deviation. The specific scheme takes the smooth fitting interval of 22s (23 points). Using the average smoothed position coordinates, a 23-point linear fitting is performed point by point to obtain the first derivative to obtain the speed, and the second derivative is obtained by four fits to obtain the acceleration. Wind speed can be calculated based on this calculation.

Considering that the average step size along the separation distance direction on different detection is different, but the 20s' smooth will filter out the noise and small turbulence of 100m to 200m. In this paper, the gravity wave scale is about 5km, which comes from the statistical characteristics of the structural function calculation. We admit that different smoothing methods will cause differences in wind speed trends, which will bring about changes in $\delta u_L(r)$. However, if all data sets are smoothed in the same way (that is, the standard is uniform), the internal comparison will not be affected by different smoothing points (provided that the smoothing points are not too large to affect the gravity wave scale).

2) L95: How exactly is defined which data are used and which not (i.e., what is "several hundred meters")? This might possibly have an impact on the results, as only the cases with certain atmospheric conditions are taken. How do the results change if these limitations are set to be for example stricter?

**Response**: We now take the data from Wuhan on October 23 as an example to explain how to select the flat-floating data:

The variation of X, Y and Z coordinates, and horizontal two-dimensional trajectories of the radiosonde with time are shown in Figure R1a–d. The X direction is the east direction, the Y direction is the north direction, and Z is the flat-floating height. The zonal movement distance is much larger than the meridional movement distance. The synthesized horizontal motion trajectory is shown in Figure 1Rd. If we want to extract the characteristics of the gravity wave from the data obtained by the airborne flat-floating sounding system, we need to first select the appropriate data segment. Here our selection principle is as follows: the data segment for analysis must satisfy that the data along separation distance direction is monotonic. For example, the solid line part in Figure R1d represents the data of the horizontal segment selected for the analysis by this sounding data. Then, X direction (zonal) is taken as the separation distance direction. The other datasets are processed in the same way. On this basis, ensure that the level fluctuation is within a few hundred meters, as shown in the red rectangle in Figure R1c.

In fact, the flat-floating segment we exclude here can not be used for the study of horizontal data. So, we're not excluding part of the atmospheric conditions, but we're excluding the period when the flat-floating attitude is not ideal. This is because even if this part is retained, it cannot accurately reflect the atmospheric conditions in the quasi-horizontal direction. These discarded segments are mainly due to sudden increase in altitude or sudden decrease in altitude in a very short period. The reason is more related to the flight state of the instrument itself, rather than from changes in the atmosphere. For example, after the outer ball explodes in the early stage of flat flight, it suddenly drops in altitude, or after flat-floating for several hours, the pressure difference between inside and outside the balloon is too large (the balloon is not closed), and it cannot maintain quasihorizontal movement.

For example, the part outside the red rectangle in Figure R1c is the noneffective flight segment, so it is excluded. And we did the same thing with all the data sets. Some detections do not retain an effective flat-floating segment for a long enough time (several hours), so it is difficult to ensure that the obtained structure function has statistical characteristics, and the entire flat-floating segment is excluded.

[Figure]

**Figure R1.** The variation of (**a**) X coordinate, (**b**) Y coordinate and (**c**) Z coordinate with time; and (**d**) the trajectory of the airborne radiosonde in the XOY plane, where the solid line is the data segment selected for subsequent analysis.

As for your comment "How do the results change if these limitations are set to be for example stricter", to illustrate this, the results of the perturbation parameters calculated for a slightly stricter limitation (red rectangle, reference fluctuating height) is show:

[Figure]

**Figure R2. (a)** Multi-order structure function, **(b)** third-order structure function (the red dots represent negative values), **(c)** multi-order singular measure, and **(d)** slope K(q), calculated from reference limitation (red rectangle)

And a looser limitation which contains noneffective flight segment (orange rectangle, larger fluctuating height) is shown:

[Figure]

**Figure R3. (a)** Multi-order structure function, **(b)** third-order structure function (the red dots represent negative values), **(c)** multi-order singular measure, and **(d)** slope K(q), calculated from looser limitation (orange rectangle)

And a much stricter limitation (black rectangle, smaller fluctuating height) is show:

[Figure]

**Figure R4. (a)** Multi-order structure function, **(b)** third-order structure function (the red dots represent negative values), **(c)** multi-order singular measure, and **(d)** slope K(q), calculated from stricter limitation (black rectangle)

Comparing Figure R2, R3, and R4, it can be seen that the shape distribution of multi-order structure function, third-order structure function, multi-order singular measure, and slope K(q) are basically the same, even under different height fluctuation limitation. In addition, the direction of the energy cascade is also consistent, indicating that the evolution state of the small-scale gravity wave does not change with the change of the limitations, which once again proves the robustness of the statistical results obtained by using the structural function. The change in C1 is even more subtle, even if the height fluctuation increases by a few hundred meters (from the black rectangle to the orange rectangle), the change in C1 is only 0.02. The change in H1 is a little more obvious, with a difference of 0.06. The reduction of H1 in Figure R4 comes from a decrease of the larger inclination in the black rectangular box (This has been explained in section 3.2 of He et al., 2022, https://doi.org/10.1007/s00376-021-1110-2).

In addition, Figure 7 in our manuscript also shows that the change of height is closely related to H1, so if the threshold of height is significantly changed in the selection of quasi-horizontal motion, the value of H1 will also change. Then, in order to exclude the change of H1 affected by height, we only discussed the relationship between C1 and other parameters, because C1 has nothing to do with the change of height.

In fact, we will also mention in the later reply that the change of C1 and H1 mainly comes from whether they pass through different physical flow regions, because the calculation of the parameters mainly comes from the change of the horizontal wind field. Of course, in order to ensure the quasi-horizontal movement, we also exclude the part of the height in a short time (tens of seconds to a few minutes) sharp rise and fall (up to

hundreds of meters) when we manually screen the flat-floating section.

3) L106: Can the rising and flat-floating stages be considered to contain concurrent effects? I am thinking whether the conditions could change during the rising or flat-floating motion. How long is the rising stage?

**Response**: We now take the data from Wuhan on October 23 as an example to explain rising and flat-floating stages:

[Figure]

**Figure R5.** (**a**) Variation in altitude (blue curve), slope distance (red curve) with time and (**b**) variation in vertical ascend rate with time. Two vertical dashed lines are used to distinguish the three stages of "rising, flat-floating and falling".

The ascending movement is approximately uniformly with time, and the average vertical speed is 5.28 m/s. After the outer ball explodes, there is a short sudden drop in height with time, which is a process of finely controlling the residual amount. Afterward, the atmospheric buoyancy–gravity balance is achieved at a predetermined height, leading to a stable long-time flat-floating. The average vertical speed during flat-floating is 0.02 m/s. The variation of altitude with time is basically small. When the inner balloon and the parachute begin to separate, the descent phase starts, and the radiosonde starts free-falling. After the parachute is opened, the initial fall speed is large, and the maximum fall speed can reach 25 m/s. Under the resistance of the parachute, the falling speed begins to gradually decrease, and the system moves from the low-density area to the high-density area. Until the radiosonde drops near the lower stratosphere, the falling speed is approximately the same as the rising speed, and the descending state of the system tended to be stable.

The duration of the rising stage is usually between 1 and 1.5 hours, which can also be seen from Figure A1. In addition, the XY trajectories of the ascending and drifting phases are plotted as follows:

[Figure]

**Figure R6.** The trajectory of the airborne radiosonde in the XOY plane, where the red line is the rising stage and the blue line is the flat-floating stage.

And you can see that the distance in the whole horizontal direction is close to 200 kilometers. In other data sets, from the initial point of the basic ascent stage to the end position of the flat-floating stage, the length of the horizontal plane along the separation distance direction is also within a few hundred kilometers. This paper mainly discusses the relationship between the small-scale gravity waves in the stratosphere and the inertial gravity waves. Inertial gravity waves, on the other hand, typically have horizontal wavelengths ranging from hundreds to thousands of kilometers and can be effectively captured by radiosondes.

So, we think that the rising and flat-floating stages be considered to contain concurrent effects.

4) L117: I have no experience with such an analysis but from an intuitive point of view, I have a problem with the definition of the separation direction. If the trajectories were mostly zonal or meridional, I would understand that it makes sense to take this direction. The trajectories depicted in Figure 1 are, however, often in some angle that does not seem to be parallel with either of these directions, so taking the projection to them will modify the multi-order structure function. Did you consider taking some fitted direction of the trajectory as the separation direction?

**Response**: Yes, we have also considered the way you mentioned, that is, after linear fitting according to the flat drift trajectory, the fitting direction is taken as the separation distance direction. Let's take the YC case on October 15$^{th}$ in Figure 3 as an example to illustrate this method.

First, the trajectory of flat-floating stage, and the variation of X coordinate, Y coordinate, and Z coordinate during flat-floating stage are plotted as follows:

[Figure]

**Figure R7. (a)** The trajectory in the XOY plane, and the (b) X coordinate, (c) Y coordinate, and (d) Z coordinate during flat-floating stage.

In order to ensure straight line fitting, it is necessary to screen the flat drift trajectory and select the region that can be approximated as a straight line for linear fitting (as suggested by the reviewer). The selected period is represented by the red rectangular box in Figure R7. Then the data part that can be processed by line fitting is obtained in Figure R8.

At this time, the flat floating balloon is moving approximately in a quasi-straight line in the northwest direction. Therefore, it is necessary to transform the original zonal and meridional wind components into the directions parallel ($u_L$) and perpendicular ($u_T$) to the separation distance through coordinate transformation, as shown in Figure R8.

At this point, the original XOY coordinate system is rotated parallel to the linear fitting direction, and the separation distance and wind velocity components $u_L$ and $u_T$ are calculated in the new coordinate system.

[Figure]

**Figure R8. (a)** The trajectory in the XOY plane, and the (b) X coordinate, (c) Y coordinate, and (d) Z coordinate for line fitting section.

[Figure]

**Figure R9.** (a) Multi-order structure function, (b) third-order structure function, (c) multi-order singular measure and (d) slope K(q) obtained from Yichang site at October 15 pm, and (e) multi-order structure

function, (f) third-order structure function, (g) multi-order singular measure and (h) slope K(q) obtained from Yichang site at November 10 pm.

Then, we calculated the disturbance parameter results under the new method, as shown in Figure R9, the figure also contains Yichang case at November 10 pm. Clearly we found some differences between the post-fitting and pre-fitting results.

Take Yichang case at November 10 pm as an example to illustrate the difference between the results before and after fitting. The result is shown in Figure R10.

[Figure]

**Figure R10.** (a) The trajectory in the XOY plane before fitting, (b) The trajectory in the XOY plane within the fitting segment, (c) third-order structure function before fitting, (d) third-order structure function within the fitting segment, (e) the longitudinal (along the path) velocity before fitting, (f) the longitudinal (along the path) velocity within the fitting segment, (g) the transverse (normal to the path) velocity before fitting and (h) the transverse (normal to the path) velocity within the fitting segment obtained from Yichang site at November 10 pm.

It can be clearly seen from Figure R10 that the third-order structure functions of the two are significantly different before (left panel) and after (right panel) fitting. The reason for this is that in the process of linear fitting, in order to ensure the approximate straight line of the trajectory, partial curves and trajectories that deviate significantly from the straight line are omitted (this treatment is used in all data sets). According to equation 1 in the manuscript, the inconsistency between the convergence and divergence of velocity on adjacent scales leads to internal instability. The balloon itself moves with the wind, so when there is a sudden change in the velocity field, the trajectory of the flat drift will naturally

change. For example, in Figure R10a, the balloon shifts from moving north to moving south near X=40km, which is caused by changes in wind field fluctuations, and can reflect the instability of atmospheric disturbance. When linear fitting is carried out, in order to ensure that the longitudinal velocity is basically along the moving direction, this part of the trajectory needs to be omitted, as shown in Figure R10b. In addition, after this treatment (linear fitting), the omitted part also corresponds to the large fluctuation region of the wind field (Figure R10e-h).

In Lu's study (2008), they also pointed out that the two scales (for example, inconsistency in the direction of the energy cascade in Figure R10c) are related to different physical flow regimes. In balloon observations, this different physical flow regimes will be represented by curved (non-linear) trajectories. Therefore, in order to retain this recognition of different physical flow regions, we chose zonal or meridional projection (which can decompose the curved trajectory into zonal or meridional).

However, with your encouragement and suggestion, we decided to supplement the information in the straight line part. Here we compare the statistical results of the two processing methods. These two data sets can also reflect two scenarios: the scenario that includes the direction change of wind component (before fitting, zonal or meridional projection was adopted) and the scenario that does not include the direction change of wind component (within the fitting segment, the separation distance follows the direction of the fitted line). In simple terms, it is divided into the case that covers all physical flow regimes (before linear fitting) and the case that only considers a single physical flow regime (after linear fitting).

According to your suggestion, we have made the following changes:

L329, A new section is added: 4.6 Calculation for a single physical flow regime.

5) L166: Discarding some more cases makes me suspicious, these things could really bias the results. Did you study why K(1) is not close to 0 in these cases? Are the trajectories somewhat special? If you make some change to the analysis (for example, taking another separation direction), does this change?

**Response**: Yes, thank you for your question. Allow me to explain the problem from two aspects:

a.  Based on mathematical formula.

Based on the mathematical formula 9, to obtain the value of C1, the numerator and denominator must both approach 0 as q approaches 1, thus obtaining the value of C1 according to the L'Hôpital's rule. The derivation of C1 is given in detail in Marshak et al, 1997.

b.  Analysis based on specific cases.

Let's take WH case at October 18 pm as an example to illustrate it. In this example, the calculated K(1) is 0.04.

The result of the flat-floating trajectory is shown in Figure R11. This is projected in the meridional direction. It should be noted that the two red rectangles in Figure R11a are dominated by the zonal displacement. In Figure R11c, the periods with small Y displacement over time are the periods between 0-593s and 9111-11500s, respectively.

We also plot the change of zonal and meridional winds over time, as shown in Figure

R12. In this case, the longitudinal velocity ($u_L$) is the meridional wind component (v). According to the formula 5-9, the larger value of K(1) is mainly due to the larger ensemble average value of the velocity increment (that is, the ensemble average of the longitudinal velocity difference over the separation distance r on the whole data).

[Figure]

**Figure R11.** (a) The trajectory in the XOY plane, and the (b) X coordinate, (c) Y coordinate, and (d) Z coordinate during flat-floating stage.

[Figure]

**Figure R12.** (a) zona wind component, and (b) meridional wind component

In Figure R12, the longitudinal velocity increment in the two red rectangular boxes (the same time period as in Figure R11) has several segments for the sudden increase velocity, and the presence of these segments causes the $\delta u_L$ (r) value to be larger.

Due to the trajectory characteristics of this detection, it is not suitable for zonal decomposition (not satisfied with the premise of monotonically increasing or decreasing

along the zonal). However, if we discard the data segment in the red rectangular boxes and calculate along the zonal direction (this is also the way to use line fitting for this case, and the fitting direction is the black dashed line in figure R11a), we can recalculate the K(1) value to 0.004. This shows that some of the original detection trajectories are too irregular in the process of zonal or meridional projection and can not satisfy the decomposition of this direction.

[Figure]

**Figure R13.** slope K(q) obtained (a) before and (b) after linear fitting.

Of course, according to your suggestion, we have reprocessed the result in the manuscript and taken the linear fitted direction of the trajectory as the separation direction. Therefore, when calculated according to linear fitting (and eliminating irregular tracks that do not meet the quasi lines), there are no differences in K(1) due to different separation directions, and there are no abrupt regions of the wind field, which will cause anomalies in K(1).

When considering all physical flow regions (as mentioned in our previous reply), K(1) serves as a threshold to exclude the conditions that do not satisfy the statistical characteristics of the intermittent parameters in this case.

6) L215: The difference between the distributions might be just an effect of the trajectory directions being more diverse in autumn, i.e., due to the mean flow?

**Response**: Indeed, thank you for offering a possible explanation for this phenomenon. The Gradient Richardson number Ri is often used to reflect the frequency of turbulence, with Ri<0.25 indicating the occurrence turbulence. Turbulence is more affected by buoyancy frequency and vertical wind shear than by wind speed itself:

$$R_i = \frac{\overline{N^2}}{\left(\frac{\partial \overline{u}}{\partial z}\right)^2 + \left(\frac{\partial \overline{v}}{\partial z}\right)^2}$$

At the same time, viewpoints that the intermittency of turbulence, sensor performance, and regional source characteristics can lead to the deviation of turbulence peaks have also been pointed out in previous studies.

In this study, the calculation of turbulence comes from the data of the ascending section, which is plotted separately as follows:

According to your suggestion, we have revised this part:

Changed "The deviation of turbulence peaks in different studies may come from the intermittency of turbulence, sensor performance, and regional source characteristics (Ko and Chun, 2022; Zhang et al., 2019; Lv et al., 2021)"

To "The deviation of turbulence peaks in different studies may come from the intermittency of turbulence, sensor performance, mean flow, and regional source characteristics (Ko and Chun, 2022; Zhang et al., 2019; Lv et al., 2021)"

7) L229: The figures displaying the "intuitively seen" results of the wind speed disturbance do not look convincing to me. Could you argue your observations more, e.g., by highlighting the parts of the plot that should illustrate this fact? Also, I guess that the roughness of the delta u_L sequence should depend on the angle of the trajectory, as the projection to the separation direction could make for example distant points closer to each other, depending on the angle.

**Response**: Thank you for your suggestion, and our reply is as follows:

"The figures displaying the "intuitively seen" results of the wind speed disturbance do not look convincing to me. Could you argue your observations more, e.g., by highlighting the parts of the plot that should illustrate this fact?"

Yes, we redrawn the graph and marked the local areas in the data sequence that would cause intermittent parameters to be too large. At the same time, before the calculation was carried out, the data was re-inspected and quality controlled, and the segments with many drift values due to abnormal positioning systems were rounded off.

It should be noted that after ensuring that the positioning information data is not abnormal, the calculated wind speed series is free of outlier interference. The larger C1 at this time is the real transient from the actual wind field. In Figure R14, the value of H1 is related to the smoothness of the data series, that is, the denser the wave packets superimposed on the fluctuation trend, the smaller the H1. The value of C1 is related to the singularity degree of the data series. In a series composed of several wave packets in a local region, the more disturbances that deviate significantly from the mean state, the larger the C1 value.

According to your suggestion, we have made the following modifications:

Changed "It can be intuitively seen from the results of the wind speed disturbance that, the lower the H1 value, the rougher the data sequence, accompanied by more wave packets; the larger the C1, the more singular the fluctuation, accompanied by stronger disturbances deviating from the average state."

To "The value of H1 is related to the smoothness of the data series, that is, the denser the wave packets superimposed on the fluctuation trend, the smaller the H1. The value of C1 is related to the singularity degree of the data series, that is, the more disturbances that deviate significantly from the mean state in a local region, the larger the C1 value. The protruding part of the purple circle in Figure 6 is the local area of the disturbance sequence (the one with the larger C1 value) that causes the intermittent parameter to be too large. Taking two cases of GZW as examples (Figure 6), compared with the detection at October 17 pm, the detection at October 20 pm has smaller H1 and larger C1. The data series at October 20 pm is rougher with denser wave packets, and there are more obvious strong perturbations deviate from the mean state in the local area."

[Figure]

**Figure R14.** The third-order structure function (left panel) and the longitudinal velocity component perturbation (right panel) for the selected cases in the corresponding red rectangles. In order to better

compare the roughness and singularity of the velocity component, the longitudinal velocity component perturbation is used here after removing the background field by using the fourth-order polynomial fitting. The protruding part of the purple circle is the local area of the disturbance sequence (the one with the larger C1 value) that causes the intermittent parameter to be too large.

"Also, I guess that the roughness of the delta u_L sequence should depend on the angle of the trajectory, as the projection to the separation direction could make for example distant points closer to each other, depending on the angle."

Yes, you're absolutely right. The reason for our zonal or meridional decomposition (projection) here has been explained before. Under this premise, in fact, the purpose of Figure R14 is to explain the quantization value on the left panel through the detailed structure of wind speed on the right panel, so as to combine the quantization method with the physical characteristics of the study object.

The roughness of the delta u_L sequence should depend on the angle of the trajectory, when the angle is determined by the dominant direction (an angle of approximately 45° makes almost negligible difference in the disturbance value from the zonal or meridional decomposition of the line segment). in the actual sequence, the signal features that cause too large C1 values and too small H1 values can still be visually observed, but here we only highlight a few obvious local regions, and there are other regions that also contribute to the excessive C1 values.

8) L266 and further: "As KHI increases…" – I strongly disagree with this interpretation of the plots. In my opinion, these "trends" are deduced just from a few outliers. In some cases, they are not visible for both seasons, even leading to an opposite "trend" for one of the seasons than the one stated in the text (Figure 7i for autumn). To make these results plausible, I believe that it is necessary to look at the trajectories of these outliers if they are somehow special and test all the previous steps in the analysis that lead to a decrease of number of considered data – maybe, they would produce another group of outliers that might change/support the trend (none of which would support the interpretation).

**Response**: Thank for your comments, we fully agree with your suggestion and make the following modifications:

a) Deleted the place where the expression of relevance is not obvious.

Changed "As KHI increases, the horizontal wavelength of IGWs decreases (Figure 9i), while the data sequence of SGWs tends to be rougher (Figure 9g)."

To "As KHI increases, the horizontal wavelength of IGWs decreases (Figure 9i)."

b) To discuss the possibility that the maximum or minimum value of the edge region in the scatter results is caused by the wild value.

We recalculated H1 and C1, where quality control and pre-judgment were performed for each selected flat-floating segment. Considering that the calculation of wind speed comes from the coordinates of the positioning system, the pre-judgment is to observe the difference between longitude and latitude at adjacent times. If the curve has no outlier value, it indicates that the positioning system works normally in the flat drift stage, and the obtained wind speed is also credible.

The judgment method is shown in Figure R15, left panel shows the case where the positioning data is abnormal, and right panel shows the case where the positioning data is normal.

[Figure]

**Figure R15.** (a) wind velocity, (c) latitude difference, and (e) longitude difference for the case where the positioning data is abnormal, and (b) wind velocity, (d) latitude difference, and (f) longitude difference for the case where the positioning data is normal

Obviously, the difference of positioning coordinates in adjacent time can identify the abnormal situation of positioning data, that is, there are a large number of wild values in a stable increment. Even if there is a sudden increase in wind speed, the transformation of the positioning data should be continuous, and this wild value comes from the anomaly of the signal received by the positioning system. So the data in this case is discarded.

Under this premise, the presence of larger C1 values can completely eliminate the interference of outlier values.

Also, we make the following modifications:

L195 Added "Considering that the calculation of wind speed comes from the coordinates of the positioning system, it is necessary to make sure that there is no wild value interfering with the results. The difference of positioning coordinates in adjacent time can identify the abnormal situation of positioning data, that is, weather there are obvious wild values in the difference of longitude or latitude. Figure A4 shows the cases for abnormal and normal positioning data, and these abnormal cases are screened out."

Here, we redraw the specific trajectories and wind field information of the right-most values in Figure 7i to show that they are not outliers.

"As KHI increases…" – I strongly disagree with this interpretation of the plots. In my opinion, these "trends" are deduced just from a few outliers.

Please allow me to plot the three KHI cases on the far right as follows:

They are cases from Anqing site at November 14 pm, Yichang site at October 14 am, and Yichang site at October 14 pm, respectively.

[Figure]

**Figure R16.** Vertical distribution of (a) X coordinates, (b) Y coordinates, (c) zonal (blue)/meridional (red) winds, and(d) gradient Richardson numbers from Anqing site at November 14 pm.

[Figure]

**Figure R17.** Vertical distribution of (a) X coordinates, (b) Y coordinates, (c) zonal (blue)/meridional (red) winds, and(d) gradient Richardson numbers from Yichang site at October 14 am.

[Figure]

**Figure R18.** Vertical distribution of (a) X coordinates, (b) Y coordinates, (c) zonal (blue)/meridional (red) winds, and(d) gradient Richardson numbers from Yichang site at October 14 pm.

It should be noted that the calculation of KHI and the horizontal wavelength of IGW are all from the data of the rising segment (vertical direction), so the data has not been filtered like the horizontal direction did, and the data quality is generally good. The three cases selected above are from the three data points in the rectangular box in Figure R19(a). It can be seen that the excessive KHI in these three cases does not come from the wild value or abnormal trajectory (the wind speed trends and trajectories shown in the three graphs are normal and reasonable, just as other detections). Excessive KHI results from detailed structural differences in wind velocity gradients, which are caused by differences in the atmospheric disturbances behind them.

[Figure]

**Figure R19.** (a) The same as figure 7i in the manuscript, and (b) the same as figure 7f in the manuscript

Besides, we also draw the trajectories and wind fields of the three cases in which C1 value is too large in Figure 7f in the manuscript (Figure R19b). They are cases from Anqing site at November 14 pm, Yichang site at October 14 am, and Yichang site at October 14 pm, respectively.

[Figure]

**Figure R20.** (a) flat-floating trajectory, (b) latitude difference, (c) longitude difference and (d) wind speed from Ganzhou site at June 14 am

[Figure]

**Figure R21.** (a) flat-floating trajectory, (b) latitude difference, (c) longitude difference and (d) wind speed from Nanchang site at June 25 pm

[Figure]

**Figure R22.** (a) flat-floating trajectory, (b) latitude difference, (c) longitude difference and (d) wind speed from Ganzhou site at October 20 pm

Similarly, for these three cases with high C1 values, the flat-floating trajectory and wind field are normal and reasonable, and the influence of outliers can be completely excluded.

In the summary and conclusion, we also added corresponding expressions to make the discussion more reasonable and convincing:

Added "Of course, given the limited number of samples and the different perturbation extraction methods in vertical and horizontal directions, the potential connection between these multi-scale fluctuations may not be significant. However, the linear relationship between disturbance from IGW and SGW can be significant if only kinetic energy is considered (the calculation of the disturbance parameter in SGW is derived only from the wind speed), as shown in Figure A5. This also indicates that the enhancement of SGW is indeed accompanied by the weakening of IGW. Besides, regardless of whether it has been linearly fitted or not, this significant linear relationship exists."

Added "The use of RTISS provides an opportunity for related research: that is, it is possible to achieve quasi-seamless detection of the atmospheric structure from both the vertical and horizontal directions inside the stratosphere at the same time. The relatively high resolution is also conducive to better capturing the fine structure of atmospheric disturbances. Taking the inertial gravity waves and small-scale gravity waves studied in this manuscript as an example, the effective capture of different disturbances in continuous time based on RTISS on different cross-sections is impossible to achieve with other single measurement methods. Even if it is limited by the sample size and the differences in calculation methods, there may be some not completely significant relationships in the discussion of different wave types and their relationship with ozone. The exploration of stratospheric atmospheric disturbances and material transport using this new detection method is still worthy of continuous follow-up and improvement. As valid samples gradually accumulate, these relationships may become more significant and robust."

9) L287: Does it make sense to average over the area? Did you test for an example that the results are similar to averaging over the actual trajectory, which would be probably the correct but more complicated way?
**Response**: Thank for your comments, our response is as follows:

The maximum differences in latitude for most flat-floating trajectories are less than 0.25°. In the process of area averaging, there are usually only 2-3 ERA5 data really in the latitude range of the flat-floating trajectory. However, there are still some detections without matched ERA5 data in the latitude (longitude) range of the flat-floating trajectory. Therefore, for these cases without matched ERA5 data within the latitude (longitude) range covered during flat-floating stage, we extend the latitude (longitude) range to a width extending 0.25° north (east) and south (west) from the center point of the trajectory. Even so, each set of detection can match only a few ERA5 data.

I fully agree with you that if the quality of the data allows (i.e. the resolution of the reanalysis is higher, at least to the mean latitude resolution of the trajectory), it may be better to screen out the reanalysis data that matches the trajectory. However, this is not possible in practice (ERA5 has a resolution of 0.25*0.25). Thank you for this suggestion, and we would appreciate it if you could understand the difficulty in changing our matching

method.

According to your suggestion, we will revise the expression of matching method as follows:L288 Added "In the process of area averaging, there are usually only two or three ERA5 data points within the latitude (longitude) range of the flat-floating trajectory. However, there are still some cases without matched ERA5 data, we extend the latitude (longitude) range to a width extending 0.25° north (east) and south (west) from the center point of the trajectory. In this way, ERA5 data and trajectory can be matched as much as possible under the premise that there is data in the matching area."

Technical corrections:

1) L20, L74: Use commas as elsewhere.

**Response**: Thank you for your suggestion, I have modified it accordingly.

Changed "GWs are excited by wave sources in the troposphere, including topography, convection and wind shear, etc"

to "GWs are excited by wave sources in the troposphere, including topography, convection, and wind shear, etc".

changed "including wind field, temperature, air pressure and relative humidity"

to "including wind field, temperature, air pressure, and relative humidity"

L95: changed "(a) Multi-order structure function, (b) third-order structure function, (c) multi-order singular measure and (d) slope $K(q)$"

to "(a) Multi-order structure function, (b) third-order structure function, (c) multi-order singular measure, and (d) slope $K(q)$"

changed "and (e) multi-order structure function, (f) third-order structure function, (g) multi-order singular measure and (h) slope $K(q)$"

to "(e) multi-order structure function, (f) third-order structure function, (g) multi-order singular measure, and (h) slope $K(q)$"

L258: changed "The blue, red and black lines in (a)–(c) represent linear fitting results of summer, autumn and all data, respectively"

To "The blue, red, and black lines in (a)–(c) represent linear fitting results of summer, autumn, and all data, respectively'

L300: Changed "The error bar diagram of (a) intermittent parameters C1, the ozone mass mixing ratio (OMR) and potential vorticity (PV) at (b) 10hPa,"

To "The error bar diagram of (a) intermittent parameters C1, the ozone mass mixing ratio (OMR), and potential vorticity (PV) at (b) 10hPa,"

2) L22, L27, L28, L111, L156, L160 and further: Check the notation of gravity waves. GWs is defined in plural but afterwards, often (not always) GW is used for the plural form..

**Response**: Thanks to your careful check, all the places where the plural form should be used have been corrected

3) L22: Article needed before wave amplitude.

**Response**: Thanks for your comments, we make the following modifications:
Changed "During upward propagation of GWs, due to the decrease of atmospheric density and the increase of wave amplitude"
To "During upward propagation of GWs, due to the decrease of atmospheric density and the increase of the wave amplitude"

4) L23: Use instability instead of unstable.
**Response**: Thanks for your comments, we make the following modifications:
Changed "unstable" to "instability"

5) L27: Plural in general circulation models.
**Response**: Thanks for your comments, we make the following modifications:
Changed "general circulation model (GCM)" to "general circulation models (GCMs)"

6) L28: Please formulate more carefully the sentence "The scale of GW is relatively small and…" Just a part of the gravity wave spectra cannot be resolved.
**Response**: We apologize for our lack of precision and we make the following modifications:
Changed "The scale of GWs is relatively small and cannot be resolved in models with relatively rough resolution"
to "Part of the GWs have relatively small scales and cannot be resolved in models with relatively rough resolution"

7) L31: "GW parametrisation…" Add an article or switch to plural.
**Response**: Thanks for your comments, we make the following modifications:
Changed "GW parametrisation" to "The GW parametrisation"

8) L34: Use the abbreviation GWs.
**Response**: Thanks for your comments, we make the following modifications:
Changed "Gravity wave" to "GWs"

9) L43: Models instead of model.
**Response**: Thanks for your comments, we make the following modifications:
Changed "model" to "models"

10) L44: Some one-sentence explanation of the RTISS system is definitely needed at this place. The description of the three stages is confusing if I do not know that it is some kind of balloon measurement.
**Response**: Thank you very much for reminding us, we make the following modifications:
Changed "The round-trip intelligent sounding system (RTISS) is a new detection technology developed in recent years (Cao et al., 2019), which can capture atmospheric fine structure information of the troposphere and stratosphere through the three-stage (rising, flat-floating, and falling) detection."
to "The round-trip intelligent sounding system (RTISS) is a new detection technology developed in recent years (Cao et al., 2019), which can capture atmospheric fine structure

information of the troposphere and stratosphere through the three-stage (rising, flat-floating, and falling) detection. That is, the outer balloon carries the radiosonde for ascending detection, and the inner balloon continues to carry the radiosonde for stratospheric detection after the outer balloon explodes, and the radiosonde is carried by the parachute for descending detection after the flat-floating is over."

11) L52: In scientific texts, plural for data ("data used in the paper are…") might be better.
**Response**: Thank you very much for pointing out that detail, we make the following modifications:
Changed "The data used in this paper" to "data used in the paper are"

12) L53: Define the abbreviations of the sites.
**Response**: Thank you very much for pointing out that detail, we make the following modifications:
Changed "covering six sites including Yichang, Wuhan, Anqing, Changsha, Nanchang, and Ganzhou in China"
to "covering six sites including Yichang (YC), Wuhan (WH), Anqing (AQ), Changsha (CS), Nanchang (NC), and Ganzhou (GZ) in China"

13) L55: I am missing the information on the approximate number of releases in winter and summer.
**Response**: Thank you very much for pointing out that detail, we make the following modifications:
Added "There are 245 detections in autumn (34 in AQ, 34 in GZ, 46 in NC, 43 in WH, 47 in YC, and 41 in CS) and 245 detections in summer (40 in AQ, 48 in GZ, 43 in NC, 44 in WH, 50 in YC, and 20 in CS)."

14) L64: Description of the colours in Figure 1 (c-h) is missing.
**Response**: We are sorry for causing such negligence. We have replaced the graphics and added the description of colors:

15) L71: "several hours apart in the vertical direction" What does it mean? How long is the rising and falling – how many values are there?
**Response**: What we want to express is that there are a few hours between the end of the detection in the rising stage and the beginning of the detection in the falling stage, which enables encrypted observations in the vertical direction. In order to avoid misunderstanding of the meaning of the expression, we have made the following changes:
Changed "RTISS aims to maintain a relatively low cost while achieving encrypted observations (rising and falling) several hours apart in the vertical direction"
To "RTISS aims to maintain a relatively low cost while achieving encrypted observations several hours apart in the vertical direction (several hours between the end of the detection in the rising stage and the beginning of the detection in the falling stage)"
We divide all the detection data (there is no distinction for the data quality or the flat-floating effect) into three stages. The running time is divided into two categories: autumn

and summer. The quantity histogram is drawn as follows::

[Figure]

**Figure R14.** Histogram of the number distribution of detection duration for (a) rising, (b) flat-floating and falling stage.

Since the sampling frequency is 1Hz, the number of seconds that each stage lasts is the sample size. As can be seen from Figure R14, the duration of the rising stage is generally between 1-1.5 hours, and the duration of the flat-floating stage has a wide distribution range. However, if we consider the flat-floating effect and the minimum time required to capture the statistical characteristics of gravity waves, we have selected the data which has the duration during flat-floating for more than two hours (more than 8,000 samples) for further processing. The duration of the falling stage is within 1 hour, and the amount of data is relatively few because the data acquisition has stopped before it reaches the ground. The data of the falling segment is not considered in this manuscript, so it does not affect the discussion of the results.

16) L74: Define abbreviation for relative humidity (and use it at L77), RH is used further in the text?
**Response**: Thank you very much for pointing out that detail, we make the following modifications:
Changed "including wind field, temperature, air pressure, and relative humidity"
To "including wind field, temperature, air pressure, and relative humidity (RH)"
Changed "relative humidity" to "RH".
Relative humidity is only mentioned in the introduction of the instrument, we did not use this element.

17) L75: Missing article before "meteorological sensor".?

Response: Thanks for your comments, we make the following modifications:

Changed "carries the Beidou navigation system and meteorological sensor"

To "carries the Beidou navigation system and the meteorological sensor"

18) L97: "separation distance direction" is used before it is defined. I suggest using just the distance of the measured points instead of it here. In any case, it would be more suitable to state if the motion is quasi-horizontal or not.

Response: Yes, your suggestion is very helpful. I agree with this amendment very much. Thank you for your suggestion.

Changed "Along the separation distance direction, the flat-floating distance is usually tens of kilometers to hundreds of kilometers"

To "Along the measured points, the flat-floating distance is usually tens of kilometers to hundreds of kilometers (in the same height plane)"

19) L100: It is not clear to me here what kind of interval you interpolate to (temporal/spatial).

Response: Thanks for your comments, we make the following modifications:

Changed "and then re-interpolated to a uniform interval after the outlying and missing values are removed"

To "and then re-interpolated to a uniform temporal interval after the outlying and missing values are removed"

20) L108: Is the Text S2 available somewhere or is it an old reference? Also, there is no reference to S1.

Response: Yes, this is the old version of the expression, and there should be no supplementary material in the manuscript you reviewed. However, in this revision, we have added supplementary materials and re-added S1 and S2.

21) L110, L114, L120, L155, L159, L172, L174, L175, L176: The math symbols in text (r, r2, r3, q) should be in math style.

Response: Thank you very much for pointing out that detail, we checked the whole text and changed all the math symbols into math style.

22) L115: I recommend moving the sentence "The balloon trajectory…" together with the definition of r (L118) before the second paragraph of the subsection.

Response: Yes, your suggestion is very helpful. I agree with this amendment very much. Thank you for your suggestion.

Changed "Among them, $\langle . \rangle$ is the ensemble average, $r$ is the separation distance, and $\varepsilon$ is the energy dissipation rate. L and T represent the directions parallel to and perpendicular to the separation distance, respectively. The balloon trajectory during flat-floating stage is not a straight line, so we decompose it into the zonal and meridional directions, and take the direction of the longer projection distance as the separation distance direction. The raw

data is uniformly interpolated to the average step along the separation distance direction. Separation distance $r = l \times 2^n$, $n = 0,1 \dots, N$. $l$ is the average step along the separation distance direction, and $N$ is limited by the maximum data length."

To "Among them, $\langle . \rangle$ is the ensemble average, $r$ is the separation distance, and $\varepsilon$ is the energy dissipation rate. The balloon trajectory during flat-floating stage is not a straight line, so we decompose it into the zonal and meridional directions, and take the direction of the longer projection distance as the separation distance direction. Separation distance $r = l \times 2^n$, $n = 0,1 \dots, N$. $l$ is the average step along the separation distance direction, and $N$ is limited by the maximum data length. L and T represent the directions parallel to and perpendicular to the separation distance, respectively. The raw data is uniformly interpolated to the average step along the separation distance direction."

23) L117: Some interpolation already mentioned in section 2.2. Is it the same interpolation or is it something else?

**Response**: No, it is something else. The interpolation mentioned in section 2.2 is the processing we do when we perform quality control of the raw data. The original data is sampled at 1s interval, and the re-interpolation with the outlying and missing values removed is also interpolated to the time interval.

24) L117: Is the last value of n surely N and not N-1?

**Response**: Separation distance $r = l \times 2^n$, what is to be considered here is that the maximum r cannot exceed the total length of the data in the longitudinal direction, and at the same time there are enough $\delta u_L(r)$ at the maximum separation distance r for the ensemble average. In all the flat-floating data selected in the six sites, the N value we calculated is 13 or 14 (Here we use $l \times 2^N < 0.9*L$, L is the total length of the data in the longitudinal direction). The purpose of a coefficient of 0.9 is that, when N=13 or 14, the number of samples of $\delta u_L(r)$ ($r = l \times 2^N$) is thousands, which meets the statistical characteristics.

According to your opinion, we have made the following modifications:

Changed "and $N$ is limited by the maximum data length."

To "and $N$ is limited by the maximum data length (in the current data $N = 13$ or $N = 14$)."

25) L117, L118: It would be better readable if the sentences do not start with a math symbol.

**Response**: Thank you very much for pointing out that detail, we have made the following modifications:

Changed "Separation distance $r = l \times 2^n$, $n = 0,1 \dots, N$. $l$ is the average step along the separation distance direction, and $N$ is limited by the maximum data length (in the current data $N = 13$ or $N = 14$)."

To "Separation distance can be determined as $r = l \times 2^n$, for integers $n = 0,1 \dots, N$, where $l$ is the average step along the separation distance direction, and $N$ is limited by the maximum data length $L$ (in the current data $N = 13$ or $N = 14$)."

Changed "L and T represent the directions parallel to and perpendicular to the separation distance, respectively"

To "The directions parallel to and perpendicular to the separation distance is represented

by L and T, respectively"

26) L125: If I understand the equations correctly, Eq. (2) for q=3 is not equivalent to equation (3) unless δu_T is very small, which is probably not the case with your definition of separation direction (might be solved by interpolating to the fitted trajectory direction, as mentioned above). It is not clear in the paper which equation is used as the third order structure function.

Response: Thank you very much for pointing out that detail, the third order structure function of equation 1 in the manuscript is used to identify the state of the GWs and the energy cascade direction, and the $2\langle \delta u_L(r)[\delta u_T(r)]^2 \rangle$ term is added to relate to the energy dissipation rate $\varepsilon$. Equation 2 is mainly used to calculate the subsequent disturbance parameters H1 and C1.

we have made the following modifications:

L126 Added "It should be noted that Eq. (1) is used to identify the state of the GWs and the energy cascade direction, while Eq. (2) is used to calculate the subsequent disturbance parameters, consistent with previous studies (Lu and Koch, 2008; Marshak et al., 1997)."

27) L125: Explain δu_L before and omit here. Also, x is not defined.

Response: Thanks for your comments, we make the following modifications:

L126 Added "Where $0 \ll x \ll L - r$."

Deleted "Because $u_L$ is the quasi-Lagrangian measurement result in the horizontal direction, $\delta u_L(r)$ can be regarded as a position-independent statistical results"

L119 Changed "$\delta u_L (\delta u_T)$ is a data set that contains the difference in velocities with a separation distance $r$ on all grid points along separation direction (perpendicular to the separation direction)."

To "$\delta u_L (\delta u_T)$ is a data set that contains the difference in the longitudinal velocities $u_L$ (transverse velocities $u_T$) with a separation distance $r$ on all grid points along separation direction (perpendicular to the separation direction). Since $u_L$ is the quasi-Lagrangian measurement result in the horizontal direction, $\delta u_L(r)$ can be regarded as a position-independent statistical results."

28) L125: Dot after the equation.

Response: Thanks for your comments, in the previous reply, we added further explanation after the equation, so we still keep the comma here.

29) L126: "a position-independent statistical results" – either omit the article or use singular.

Response: Thanks for your comments, we make the following modifications:

Changed "a position-independent statistical results"

To "a position-independent statistical result"

30) L133: Probably something like "we define" instead of "we choose". The sentence is not well understandable.

Response: Thank you very much for pointing out that detail, we make the following modifications:

Changed "choose"
To "define"

31) L133: Why does the value of H1 have to be between 0 and 1?

**Response**: For data signals in the atmosphere, we first assume this field is scale invariant; its energy spectrum follows a power law,

$$E(k) \propto k^{-\beta} \text{ where } 1 \le \beta \le 3$$

The parameter H2 gives an estimate for the general spectral power law by using the Monin and Yaglom (1975) conversion law, which results in

$$1 < \beta = \zeta(2) + 1 = 2H(2) + 1 < 3$$

The upper bound can be obtained when the analyzed process is close to being an everywhere differentiable signal. The parameter H1, on the other hand, gives

$$0 < H_1 = H(1) = \zeta(1) < 1$$

which is the well-known self-similar (self-affine) exponent or Hurst parameter. The Hurst parameter measures the roughness (nonstationarity) of the signal in data, with 0 representing the roughest functional series, such as white noise, and 1 representing an infinite smoothness function (Marshak et al. 1997).
we make the following modifications:
Changed "Here we definechoose H1=H(1) as the Hurst index, with a value between 0-1"
To "Here we definechoose H1=H(1) as the Hurst index, which can measures the roughness (nonstationarity) of the signal in data, with a value between 0-1 (Marshak et al. 1997)"

32) L138, L139: Different meaning for symbols epsilon and l? Renaming would make it more comprehensible.

**Response**: Thank you very much for pointing out that detail, we make the following modifications:
L113: The mathematical symbol of energy dissipation rate is modified to $E$ (given that the energy dissipation rate occurs only once, the latter epsilon is used to mean something else). The $l$ used here is consistent with the previous definition and is the average step along the separation distance after re-interpolation. So the meaning of the symbol is not repeated here.

33) L149: Is C1 the same as C_1? It is not defined. And again, it is not clear for me why the values are between 0 and 1.

**Response**: Yes, this is our negligence. We have checked the whole text and all of them are unified as C1.
$D(q)$ can be used to represent the well-known non-increasing hierarchy of "generalized dimensions", which is first introduced by Grassberger (1983) and Hentchel and Procaccia (1983) with dynamical systems and strange attractors in mind. For $D(q) \equiv 1$, the fluctuation demonstrates a monoscaling. Otherwise, $D(q) < 1$ for q>0.
For actual wind field data, we are dealing with singular (hence skewed) $\varepsilon(\eta; x)$ distributions, so $D(1) < 1$ (In reality, there is no ideal monoscaling fluctuation). Because we try to use as

few multifractal parameters as possible to capture some physical properties of a fluctuation, only D(1) is considered, similar to the use of H (1) for a smoothness measure.

D(1) is called the information dimension and it's a non-negative value.

As a result, C1=1-D(1) has a value between 0 and 1.

Reference:

Hentschel, H. G. E., and I. Procaccia, 1983: The infinite number of generalized dimensions of fractals and strange attractors. Physica D, 8, 435–444.

Grassberger, P., 1983: Generalized dimensions of strange attractors. Phys. Rev. Lett. A, 97, 227–330.

we make the following modifications:

Changed "C1 is an intermittent parameter with a value between 0-1, reflecting the singularity of the fluctuation"

To "C1 is an intermittent parameter with a value between 0-1, reflecting the singularity of the fluctuation (Marshak et al. 1997)"

34) L166: "discard" instead of "discarded some".

**Response**: Thanks for your comments, we make the following modifications:

Changed "thereby discarded some unsatisfactory cases"

To "thereby discarded unsatisfactory cases"

35) L169: Capital T.

**Response**: Thanks for your comments, we make the following modifications:

Changed "therefore"

To "Therefore"

36) L171: Change "to illustrate" to "we illustrate".

**Response**: Thanks for your comments, we make the following modifications:

Changed "to illustrate"

To "we illustrate"

37) L173: Article before "third-order structure function".

**Response**: Thanks for your comments, we make the following modifications:

Changed "third-order structure function"

To "the third-order structure function"

38) L173: "from the third-order structure function" probably accidentally twice.

**Response**: Thanks for your comments, we make the following modifications:

Changed "a downscale energy cascade (from large to small scales) can be seen from the third-order structure function"

To "a downscale energy cascade (from large to small scales) can be seen"

39) L173: an r^3 slope

**Response**: Thanks for your comments, we make the following modifications:

Changed "a $r^3$"
To "an $r^3$"

40) L178: I don't understand how the GW scale is quantified. According to the previous section, I thought that R_w should be < 5 km?

**Response**: Sorry for this gross oversight, but the threshold set in our program is 6km. The scale of gravity waves comes from a separation distance r less than 6 km and closest to 6km. The statistical characteristics of H1 and C1 at the corresponding scale were used as the quantization of perturbation parameters of small-scale gravity waves at this scale, which was applied in earlier studies.
we make the following modifications:
L159 Changed "and the separation distance closest to 5 km (< 5 km)"
To "and the separation distance closest to 6 km (< 6 km)"

41) L180: Consider unifying the axis label style in Figure 2 a – d (changing for example 10^5 to 5 or vice versa).

**Response**: According to your suggestion, we have redrawn the graph and modified the axis label style in Figure 2c:

[Figure]

42) L180: "The red dots represent negative values" – does this mean that the plot actually displays -S3? Would be more understandable for me in the axis label.

**Response**: It is just a coincidence that the points on each graph (different separation distances r) are negative, but in b and f in Figure 3, there are positive and negative values in different separation distances, so we can use red dots and blue dots to represent the energy cascade direction on different scales at the same time, and the ordinate is actually the absolute value.

43) L181: What are the dashed lines in Figure 2d?

**Response**: The red dashed line is K(q)=0, and the blue dashed line is the fit slope at K(1). we make the following modifications:

L182: Added "In Figure 2d, the red dashed line is K(q)=0, and the blue dashed line is the fit slope at K(1)".

44) L182: Figures 2e and 2d are, to my understanding, not really connected to the remaining subplots in Figure 2 and they are even first referenced after Figure 3. I would consider moving them to a separate figure.

**Response**: Thanks for your comments, we make the following modifications:

We split this graph into two graphs, Figure 2 and Figure 3. The description of the figure in the manuscript was also modified.

45) L184: Either "an unstable GW" or "unstable GWs".

**Response**: Thanks for your comments, we make the following modifications:

Changed "Figure 3 shows cases for unstable GW and the coexistence of GW and turbulence."

To "Figure 3 shows cases for unstable GWs and the coexistence of GWs and turbulence."

46) L185: The notation of H1 and C1 in brackets is not defined before, it doesn't have to be completely clear.

**Response**: Thanks for your comments, in fact, in L178 we have proposed to use (H1,C1) to quantify. But it's probably not explicitly stated here, so with your suggestion, we make the following modifications:

Changed "The case for Yichang data at October 15 pm can be identified as an unstable GW, and the GW is quantified as (0.59, 0.10), with a scale of 5.1 km."

To "The case for Yichang data at October 15 pm can be identified as an unstable GW, with a scale of 5.1 km. The GW is quantified as (0.59, 0.10), where the first value is H1 and the second value is C1."

47) L186: Change "coexist" to "coexisting".

**Response**: Thanks for your comments, we make the following modifications:

Changed "coexist"

To "coexisting"

48) L199 – L217: I have the feeling that these paragraphs do not fit into the subsection (not about disturbance parameters)..

**Response**: Thanks for your comments, we make the following modifications:

We move this paragraph to the introduction of analytical methods in Section 3:

3.3 IGWs and turbulence parameter

Based on the data during the rising stage, we use hodograph analysis to extract IGW parameters (Bai et al., 2016; Huang et al., 2018), with a height interval of 18–25 km, thereby obtaining parameters including vertical wavelength, horizontal wavelength, intrinsic frequency, propagation direction (anticlockwise from y axis), kinetic energy,

potential energy, and momentum flux. In order to eliminate the error caused by the random movement of the balloon, the data is uniformly interpolated to an interval of 50 m. The total energy is the sum of kinetic energy and potential energy.

Based on Thorpe analysis (Ko & Chun, 2022; Thorpe, 1977; Wilson et al., 2011), the atmospheric turbulent layer is identified from the sorted potential temperature profile, thereby obtaining turbulence parameters including Thorpe length, turbulent layer thickness, turbulent kinetic energy dissipation rate, and turbulent diffusion coefficient. Optimal smoothing and statistical tests are used to distinguish "overturn" caused by real turbulent motion and artificial "inversion" caused by instrument noise and balloon motion (Wilson et al., 2011).

49) L205: "distinguish between".
**Response**: Thanks for your comments, we make the following modifications:
Changed "distinguish"
To "distinguish between"

50) L220: Missing description of colours in Figure 4.
**Response**: Thanks for your comments, we redrew the graphic and added the color description.

51) L226: "no matter whether".
**Response**: Thanks for your comments, we make the following modifications:
Changed "no matter"
To "no matter whether"

52) L227: Maybe change "with lower latitude" to "at lower latitude"?
**Response**: Thanks for your comments, we make the following modifications:
Changed "with lower latitude"
To "at lower latitude"

53) L242: Wrong description of Figure 6, same as for the previous figure.
**Response**: Thanks for your comments, we make the following modifications:
Changed "Figure 6. The atmospheric disturbance parameters (H1, C1) and the corresponding average flat-floating height (scaled to 1/40) obtained over the six sites in summer (left panel) and autumn (right panel), the mean and standard deviation of H1 and C1 are marked in blue and yellow, respectively."
To "Figure 6. the third-order structure function (left panel) and the longitudinal velocity component perturbation (right panel) for the selected case in the corresponding red rectangles. In order to better compare the roughness and singularity of the velocity component, the longitudinal velocity component perturbation is used here after removing the background field by using the fourth-order polynomial fitting"

54) L253: Three spaces between "between" and "H1".
**Response**: Thanks for your comments, we cut out two of those spaces.

55) L254: Delete the backslash symbol.

**Response**: Thanks for your comments, we deleted the backslash symbol.

56) L260: Would be more comprehensible if you state here you are writing about variables from figures 7 d – f.

**Response**: Thanks for your comments, we make the following modifications:

Changed "Due to the limitations of the sample size and the different detection objects, the linear correlation between these variables may not be statistically significant"

To "Due to the limitations of the sample size and the different detection objects, the linear correlation between these variables from figures 7 d – f may not be statistically significant"

57) L264: I would prefer the sentence "Considering…" to be reformulated. For example, "Next, we consider that (…) the Kelvin Helmholtz instability. The ratio of (…) representing the instability is used to explore (…)".

**Response**: Thank you very much for pointing out that detail, we make the following modifications:

Changed "Considering that the wave disturbance in the stratosphere is likely to be related to KHI (He et al., 2020b; Lu and Koch, 2008), here the ratio of $0 < Ri < 0.25$ between 15–25 km is used to represent the Kelvin- Helmholtz instability (KHI) to explore its connection with atmospheric disturbances."

To "Next, we consider that the wave disturbance in the stratosphere is likely to be related to the Kelvin Helmholtz instability (He et al., 2020b; Lu and Koch, 2008). The ratio of $0 < Ri < 0.25$ between 15 and 25 km representing the instability is used to explore its connection with atmospheric disturbances."

58) L264: Explain KHI here + add an article.

**Response**: Thanks for your comments, we have made changes in the last reply."

59) L265: "between 15 and 25 km" might be more understandable.

**Response**: Thanks for your comments, we have made changes in the last reply.

60) L276: Small o.

**Response**: Thanks for your comments, we make the following modifications.

61) L278: Regarded.

**Response**: Thanks for your comments, we make the following modifications:
Changed "regard"
To "regarded"

62) L280: Use "aim" instead of "hope".

**Response**: Thanks for your comments, we make the following modifications:
Changed "hope"

To "aim"

63) L284, L285: What do the words "basically" and "exactly" mean here? I am confused. Does it mean that the release is done approximately at let's say 23 UTC, so that it arrives upward at exactly 00 UTC? Please reformulate this part. Also, I believe that this information should be rather in the data or methodology part and not in results.

**Response**: Yes, you are absolutely right. The release is done approximately at 23UTC (7:00 Beijing time) and 11UTC (19:00 Beijing time). Taking into account the rise time of nearly 1–1.5 hour, it arrives upward at approximately 00 UTC and 12UTC for flat-floating detection.

we make the following modifications:

L62 Added

"In order to explore the correlation between RTISS data and atmospheric composition, we obtained ozone and potential vorticity from ERA5 reanalysis data. The release time of flat-floating detection is divided into two periods, morning and evening. The release is done approximately at 23UTC (7:00 Beijing time) and 11UTC (19:00 Beijing time). Taking into account the rise time of nearly 1–1.5 hour, it arrives upward at approximately 00 UTC and 12UTC for flat-floating detection. Therefore, the 00UTC and 12UTC data provided by ERA5 can be well combined with the observation results of RTISS in the flat-floating stage for analysis."

L282 Changed "Based on the ERA5 reanalysis data, the ozone mass mixing ratio (OMR) and PV at different pressure layers that matched the detection are selected. The release time of flat-floating detection is divided into two periods, morning and evening. The release time is basically 23UTC (7:00 Beijing time) and 11UTC (19:00 Beijing time). Taking into account the rise time of nearly 1 hour, the data of the flat-floating period exactly corresponds to 00UTC and 12UTC of ERA5. Then according to the latitude and longitude range covered by RTISS during flat-floating stage, the OMR and PV obtained from the ERA reanalysis data are averaged in the corresponding area. The matching results of different air pressure layers (200hPa, 175hPa, 150hPa, 125hPa, 100hPa, 70hPa, 50hPa, 30hPa, 20hPa, 10hPa, 5hPa, 3hPa, 2hPa, 1hPa) are calculated."

To "Based on the ERA5 reanalysis data, the ozone mass mixing ratio (OMR) and PV at different pressure layers that matched the detection are selected. According to the latitude and longitude range covered by RTISS during flat-floating stage, the OMR and PV obtained from the ERA reanalysis data are averaged in the corresponding area. The matching results of different air pressure layers (200hPa, 175hPa, 150hPa, 125hPa, 100hPa, 70hPa, 50hPa, 30hPa, 20hPa, 10hPa, 5hPa, 3hPa, 2hPa, 1hPa) are calculated"

64) L290: What is the height range where small-scale GWs are detected?

**Response:** The height of flat-floating is mainly between 20-30 km. After recalculating H1 and C1 (again through data quality control and flat-floating segment screening), the pressure layer above (10 hPa) and the pressure layer below (100 hPa) are selected to discuss the correlation between ozone and C1.

According to your suggestion, we make the following modifications:

Changed "The pressure layers selected here correspond to the height above and below the

flat-floating interval, in order to distinguish them from the height range where small-scale GWs are detected."

To "The pressure layers selected here correspond to the height above (10 hPa) and below (100 hPa) the flat-floating interval (20~30 km), in order to distinguish them from the height range where small-scale GWs are detected."

[Figure]

**Figure 8. The error bar diagram of (a) intermittent parameters C1, the ozone mass mixing ratio (OMR), and potential vorticity (PV) at (b) 10hPa, (c) 100hPa pressure layers in summer (S) and autumn (W), showing a total of 12 clusters over the six sites. The blue, yellow, and black annotations marked at the top of the subgraph indicate the Pearson correlation coefficient and significance level for OMR versus C1, PV versus C1, and OMR versus PV, respectively. Outside the brackets is the correlation of the average values of the 12 clusters (12 values), inside the brackets is the correlation of all cases of the twelve clusters.**

65) L310: Is correlation coefficient 0.5 from 12 values so significant? How large is the p-value for these levels?

**Response**: These 12 values are the average of all valid summer and autumn detections over each site, and the correlation between the average values is much larger than the total data. Since the flat-floating height is between 20km and 30km, the corresponding pressure layer is near the 10hPa~50hPa pressure layer, and the 5hPa and 3hPa at higher altitudes, as well as the 125-150hPa at lower altitudes, are enough to reflect the relationship between the two outside this altitude region. So only 10 pressure layers remain in the redrawn figure.

p-value of the 12 clusters for these levels with correlation coefficient more than 0.5 shown as follow:

5hPa, p=0.01; 10 hPa, p=0.03; 20 hPa, p=0.05; 30 hPa, p=0.07; 100 hPa p=0.08

According to your suggestion, we marked the p-value with a significant correlation

coefficient in the figure.

66) L321: "that is closely related"?.
**Response**: Thanks for your comments, we make the following modifications:
Changed "Ozone transport that closely related to the SGW occurs between 100hPa and 10hPa"
To "Ozone transport is closely related to the SGW between 100hPa and 10hPa"

67) L328: Regarding the right part of the figure, unfortunately, I don't understand it at all. Could you perhaps consider supplementing some description to the red arrows? On the other hand, I really like the left part of the figure - it is very nice and illustrative and could be very useful for an introductory (methodology) part of the paper.
**Response**: Thank you very much for your recognition of the left part of our picture. According to your valuable suggestions, we have drawn the left part separately and put it in the introductory (methodology) part.
L84: Changed "The detection principle is simply summarized as follows: in the rising stage…"
To "The three-stage detection process by RITSS described in Figure 2. In the rising stage…"

[Figure]

**Figure 2. The three-stage detection process by RITSS**

In addition, we have regrouped Figure 9 and right part of Figure 10. It can better match the negative correlation of ozone below and the positive correlation above with the activity of small-scale gravity waves.

[Figure]

Mechanism Diagram of Ozone Transport and energy transfer

**Figure 11. The mechanism diagram of ozone transport and energy transfer. Right part shows the vertical distribution of correlation coefficient between the OMR and C1 in summer and autumn (a total of twelve clusters) over the six sites at different pressure layers. When the correlation for OMR versus C1 of the average values of the 12 clusters (12 values) and of all cases are both statistically significant (p < 0.1), it is considered that the small-scale GW disturbance is closely related to the change in ozone concentration on the corresponding pressure layers, otherwise the correlation coefficient is set to 0. For the pressure layers with significant correlation coefficient, the significance level p value corresponding to the 12 clusters is marked in the figure.**

L309 Added "The mechanism diagram of ozone transport and energy transfer is shown in Figure 11. The significant positive (negative) correlation between C1 and ozone concentration in the lower (middle) stratosphere further support the argument that SGW may affect the vertical transport of ozone (right part of Figure 11). The stratospheric SGWs detected here are closely related to KHI, and previous studies have also confirmed this (He et al., 2020b; Lu and Koch, 2008). The transport capacity of IGWs on ozone is weakened due to the critical layer filtering during its upward propagation. In contrast, the high-frequency SGWs can propagate to higher altitudes (Dong et al., 2023). Ozone transport is closely related to the SGWs between 100 hPa and 10 hPa, corresponding to the weakening of IGWs in the lower stratosphere (100hPa) and the enhancement of SGWs excited by KHI. SGWs with higher phase velocities would propagate upward without encountering critical level and thus complete the ozone transport to the middle stratosphere (10 hPa) (Heale and Snively, 2015; Li et al., 2020; He et al., 2022b). The enhanced intermittency is accompanied by the weakening of IGW energy below, which also reveals the possible energy transfer from large-scale to small-scale waves."

68) L344: Add an article before "wave dissipation".

**Response**: Thanks for your comments, we make the following modifications:
Changed "wave dissipation"
To "the wave dissipation"

69) L357: Appendix is not referenced in the paper..

**Response**: Thanks for your comments, we make the following modifications:
L95 Added "The variation of the height during the whole process of RTISS over time is shown in Figure A1"

At the end, Authors are grateful to the anonymous reviewer for providing valuable comments to improve the manuscript up to this level. We greatly appreciate the time and effort you put into improving the quality of my manuscript, and we have benefited immensely from your selfless comments and suggestions. Besides, if you have more suggestions or comments about my manuscript or the content of the reply, I will always be pleased to make timely replies and revisions and benefit from communicating with you. Finally, thank you again from the bottom of my heart.

**In addition, the author also checked the full text, revised some grammar and details, and they can all be found with "track changes".**

---

## Author Comment (AC2)

The editorial support team
*Atmospheric Chemistry and Physics*
October 15th, 2023
Subject: Revision of manuscript egusphere-2023-1608

Dear Editors and Reviewers:
Thank you for your letter and for giving us the opportunity to revise our manuscript on "Identification of stratospheric disturbance information in China based on round-trip intelligent sounding system" [Paper # egusphere-2023-1608]. We have carefully reviewed the comments and have revised the manuscript accordingly. Our responses are given in a point-by-point manner below. Changes to the manuscript are shown in the revised manuscript with "track changes".
Sincerely,
Yang He
E-mail: heyang12357@sina.com

Corresponding author: Zheng Sheng
E-mail: 19994035@sina.com

**Response to Reviewer #2:**
General comments:
This study applies the structure function and singular measure to quantify the stratospheric small-scale gravity wave (SGW) over China by Hurst parameter and intermittency parameter, and discuss its relationship with inertia-gravity wave (IGW). The introduced observation system for floating balloon measurements is an interesting novel option to invetigate atmospheric disturbances in the stratosphere. Although the authors' dataset and observation could be of very high scientific value, the study in its present form suffers from several flaws and I recommend publication with suitable revisions.
Response: Thanks for appreciating our contribution and performing such an insightful and detailed review. We have made targeted revisions and replied in accordance with the specific opinions you give later. Your professional opinions are very valuable for improving the quality of our manuscript. We also look forward to your feedback on our improved work.

Specific comments:
1) The dynamics of a sounding platform and its response to atmospheric motions is of necessary knowledge before interpreting atmospheric disturbances parameter such as structure functions and intermittency parameter.
In the present case, the expected behavior of the balloon in the flat-floating phase remains unclear. From the introduction (L84-89), it is implied that this phase is characterized by quasi-horizontal motion similar to superpressure balloons (Hertzog et al., 2002; Boccara et al., 2008). Whether their detection principles are consistent? the authors should further explain the uniqueness of the detection system, and strongly recommend that the dynamic

process of detection be further elaborated.

**Respon**se: Thanks for your comments, in fact, the balloons we use are zero-pressure balloons, which are not the same as overpressure balloons.

Zero-pressure balloon is made of low-temperature polyethylene film, the ball itself is not closed, gravity and buoyancy are balanced during flat-floating stage, the bottom of the exhaust pipe so that the ball inside and outside the pressure difference is basically zero, the flight time is short (several hours). The overpressure balloon sphere is closed, and the structure of the sphere is similar to that of the pumpkin shape. The volume of the sphere is basically unchanged, and the flight time is longer (several weeks).

the detection principle of RTISS in three stages is given below

In the ascending phase, the balloon is driven without power in the horizontal direction, which can be approximated as moving with the wind field and subjected to buoyancy, gravity and air resistance in the vertical direction(Cao et al., 2019; He et al., 2020):

$$m\frac{dw}{dt} = \rho V g - mg - \frac{\pi}{8}\rho C_p D^2 w^2$$

Where $m$ is the mass of the system, $g$ is the acceleration of gravity, $w$ is the vertical velocity of the balloon, $V$ is the volume of the outer sphere, $\rho$ is the atmospheric density, $C_p$ is the drag coefficient, and $D$ is the diameter of the outer sphere.

In the horizontal floating stage, the adaptive flat-floating process is realized by controlling an appropriate net lift force from the ground. Under this condition, the vertical force is dynamically balanced, and the balloon trajectory can be regarded as an approximate horizontal motion. The motion equations are written as:

$$F_{up} = \rho_0 g \frac{V_h P_h T_0}{T_h P_0} - mg$$

$$V_h = mg/\rho_h$$

Where $T_h$, $P_h$, and $\rho_h$ are the temperature, air pressure, and density inside the balloon at height h, $T_0$, $P_0$, and $\rho_0$ are the initial temperature, air pressure and density before the balloon is released.

In the descending stage, the radiosonde descends under the parachute, from the low-density atmosphere into the high-density atmosphere, the motion equations at this time are:

$$m\frac{dV_{ux}}{dt} = -m_{au}\frac{dV_x}{dt} - \frac{\rho S V_r \left( C_d V_x + C_l \left( k V_y - j V_z \right) \right)}{2}$$

$$m\frac{dV_{uy}}{dt} = -m_{au}\frac{dV_y}{dt} - \frac{\rho S V_r \left( C_d V_y + C_l \left( -k V_x + i V_z \right) \right)}{2}$$

$$m\frac{dV_{uz}}{dt} = m_f g - mg - m_{au}\frac{dV_z}{dt} - \rho S V_r \left( C_d V_z + C_l \left( j V_x - i V_z \right) \right)\Big/2$$

$$V_r = \sqrt{V_x^2 + V_y^2 + V_z^2}$$

Where $m$ is the total mass of the system, $m_{au}$ is the additional mass (produced by the acceleration of the parachute), and $m_f$ is the air mass discharged by the system. $V_x$, $V_y$ and $V_z$ are the velocity of the system relative to the air, $V_{ux}$, $V_{uy}$ and $V_{uz}$ are the velocity of the radiosonde itself, $C_d$ and $C_l$ are the drag coefficient and the lift coefficient, and $i$, $j$, and $k$ are the cosines of the direction vector.

Considering that the specific physical process regarding the detection process has been

given in a previously published paper, here we make the following modifications:

L89 Added: "The detection system has different working principles in the three stages, and the specific dynamic process can be referred to the previous work (Cao et al., 2019). It should be noted that the RTISS uses the zero-pressure balloon to meet the needs of low-cost business observation, which is different from the superpressure balloon (Hertzog et al., 2002; Boccara et al., 2008). For the zero-pressure balloon, the bottom exhaust pipe makes the pressure difference between inside and outside the balloon basically zero, the flight time is short (several hours). While for the super-pressure balloon, the sphere is closed, the volume of the sphere is basically unchanged, and the flight time is longer (several weeks)."

2) the authors carry out their structure function analysis in space coordinate (longitudinal distance) after interpolation, in my intuition, time is the appropriate and most commonly used coordinate in which to perform the analysis similar to a quasi-Lagrange measurement. Of course, this may be treated differently from the linear fitting mentioned in the first point, I am well aware that different methods have their own rationality and limitations, and it may be interesting if the author can compare the results of different methods here. For example, compare the results in space and time coordinates? If the data can be analyzed in time coordinates, there is no need for interpolation since the measurements were sampled regularly.

Yes, we have also considered the way you mentioned, that is, after linear fitting according to the flat drift trajectory, the fitting direction is taken as the separation distance direction. Let's take the YC case on October 15th in Figure 3 as an example to illustrate this method. First, the trajectory of flat-floating stage, and the variation of X coordinate, Y coordinate, and Z coordinate during flat-floating stage are plotted as follows:

[Figure]

**Figure R1. (a)** The trajectory in the XOY plane, and the (b) X coordinate, (c) Y coordinate, and (d) Z coordinate during flat-floating stage.

In order to ensure straight line fitting, it is necessary to screen the flat drift trajectory and select the region that can be approximated as a straight line for linear fitting (as suggested by the reviewer). The selected period is represented by the red rectangular box in Figure R1. Then the data part that can be processed by line fitting is obtained in Figure R2.

At this time, the flat floating balloon is moving approximately in a quasi-straight line in the northwest direction. Therefore, it is necessary to transform the original zonal and meridional wind components into the directions parallel ($u_L$) and perpendicular ($u_T$) to the separation distance through coordinate transformation, as shown in Figure R8.

At this point, the original XOY coordinate system is rotated parallel to the linear fitting direction, and the separation distance and wind velocity components $u_L$ and $u_T$ are calculated in the new coordinate system.

[Figure]

**Figure R2. (a)** The trajectory in the XOY plane, and the (b) X coordinate, (c) Y coordinate, and (d) Z coordinate for line fitting section.

[Figure]

**Figure R3.** (a) Multi-order structure function, (b) third-order structure function, (c) multi-order singular measure and (d) slope K(q) obtained from Yichang site at October 15 pm, and (e) multi-order structure function, (f) third-order structure function, (g) multi-order singular measure and (h) slope K(q) obtained from Yichang site at November 10 pm.

Then, we calculated the disturbance parameter results under the new method, as shown in Figure R3, the figure also contains Yichang case at November 10 pm. Clearly we found some differences between the post-fitting and pre-fitting results.

Take Yichang case at November 10 pm as an example to illustrate the difference between the results before and after fitting. The result is shown in Figure R10.

[Figure]

**Figure R4.** (a) The trajectory in the XOY plane before fitting, (b) The trajectory in the XOY plane within the fitting segment, (c) third-order structure function before fitting, (d) third-order structure function within the fitting segment, (e) the longitudinal (along the path) velocity before fitting, (f) the longitudinal (along the path) velocity within the fitting segment, (g) the transverse (normal to the path) velocity before fitting and (h) the transverse (normal to the path) velocity within the fitting segment obtained from Yichang site at November 10 pm.

It can be clearly seen from Figure R4 that the third-order structure functions of the two are significantly different before (left panel) and after (right panel) fitting. The reason for this is that in the process of linear fitting, in order to ensure the approximate straight line of the trajectory, partial curves and trajectories that deviate significantly from the straight line are omitted (this treatment is used in all data sets). According to equation 1 in the manuscript, the inconsistency between the convergence and divergence of velocity on adjacent scales leads to internal instability. The balloon itself moves with the wind, so when there is a sudden change in the velocity field, the trajectory of the flat drift will naturally change. For example, in Figure R10a, the balloon shifts from moving north to moving south near X=40km, which is caused by changes in wind field fluctuations, and can reflect the instability of atmospheric disturbance. When linear fitting is carried out, in order to ensure that the longitudinal velocity is basically along the moving direction, this part of the trajectory needs to be omitted, as shown in Figure R4b. In addition, after this treatment (linear fitting), the omitted part also corresponds to the large fluctuation region of the wind

field (Figure R4e-h).

In Lu's study (2008), they also pointed out that the two scales (for example, inconsistency in the direction of the energy cascade in Figure R4c) are related to different physical flow regimes. In balloon observations, this different physical flow regimes will be represented by curved (non-linear) trajectories. Therefore, in order to retain this recognition of different physical flow regions, we chose zonal or meridional projection (which can decompose the curved trajectory into zonal or meridional).

However, with your encouragement and suggestion, we decided to supplement the information in the straight line part. Here we compare the statistical results of the two processing methods. These two data sets can also reflect two scenarios: the scenario that includes the direction change of wind component (before fitting, zonal or meridional projection was adopted) and the scenario that does not include the direction change of wind component (within the fitting segment, the separation distance follows the direction of the fitted line). In simple terms, it is divided into the case that covers all physical flow regimes (before linear fitting) and the case that only considers a single physical flow regime (after linear fitting).

Since more flat drifting trajectories are very irregular, if you select a part that is approximately straight, you can only get a significantly smaller data segment, and many structure functions at a larger separation distance cannot be calculated. At the same time, even if the part that approximates a straight line is selected, there are actually many small fluctuations superimposed on it, which cannot be completely regarded as a straight line. At this time, interpolation at equal distances will cause errors. So our point of view is that no matter what kind of processing method, the error is unavoidable, and according to the characteristics of the flat float segment data, the decomposition of the latitude and longitude can make full use of the data.

As for processing on a time basis, I hope you could understand that we have not supplemented this practice here. One reason is that the amount of work in a manuscript is sufficient considering that, after supplementing the results of linear fitting, it is already possible to cover both the results containing different physical flow regions and the results containing only a single physical flow. Another reason is that small scale gravity waves on the spatial scale cannot be investigated with time interpolation. Of course, your suggestion is very good, and we will try it in the follow-up work.

3) L95-L97: The flight segments used for analysis should be chosen carefully in order to be quasi horizontal. For example, in part of the profile in Figure A1, after the burst of the outer balloon, the platform adjusts to its equilibrium level a few hundred meters below the burst altitude. In my understanding, the authors are nevertheless using that initial segment in their analysis, even though it is contaminated by altitude variations. I would recommend to discard it.

**Response**: We are very sorry that our presentation may have caused you to have such a misunderstanding. In fact, when selecting the data of the flat drift section, we have already omitted the period just after the end of the rising stage.

We strictly select each detection by manual screening, in order to select the segment with ideal flat-floating effect for subsequent calculation. In other word, we have already

discarded the initial segment after the burst of the outer balloon.

Thanks for your comments, to make the expression clearer, we make the following modifications:

L97 Added "It should be noted that, after the burst of the outer balloon, the platform adjusts to its equilibrium level a few hundred meters below the burst altitude (Figure A1), thus the initial segment after the burst of the outer balloon is also discarded."

4) L115-120:Using structural functions, the calculation of the longitudinal velocity (Parallel to separation distance), requires careful consideration. Why not directly use the original flat drift data? what is the purpose of decomposition? Specifically, what criteria is used for reference to decompose according to the longitude direction or the latitude direction? Can the data analysis performed after the decomposition still represent the characteristics of the fluctuation?

**Response**:

Why not directly use the original flat drift data? what is the purpose of decomposition?

Because the original flat drift trajectory is not regular, if the separation distance direction is determined by linear fitting according to the original flat drift trajectory, part of the important flat-floating segment (that is, the area with significant wind field fluctuations) will be omitted. In order to illustrate this problem, we choose an example to explain:

Let's take WH case at October 18 pm as an example to illustrate it. In this example, the calculated K(1) is 0.04.

The result of the flat-floating trajectory is shown in Figure R5. This is projected in the meridional direction. It should be noted that the two red rectangles in Figure R5a are dominated by the zonal displacement. In Figure R5c, the periods with small Y displacement over time are the periods between 0-593s and 9111-11500s, respectively.

We also plot the change of zonal and meridional winds over time, as shown in Figure R6. In this case, the longitudinal velocity ($u_L$) is the meridional wind component (v). According to the formula 5-9, the larger value of K(1) is mainly due to the larger ensemble average value of the velocity increment (that is, the ensemble average of the longitudinal velocity difference over the separation distance r on the whole data).

[Figure]

**Figure R5.** (a) The trajectory in the XOY plane, and the (b) X coordinate, (c) Y coordinate, and (d) Z coordinate during flat-floating stage.

[Figure]

**Figure R6.** (a) zona wind component, and (b) meridional wind component

In Figure R6, the longitudinal velocity increment in the two red rectangular boxes (the same time period as in Figure R5) has several segments for the sudden increase velocity, and the presence of these segments causes the $\delta u_L$ (r) value to be larger.

Due to the trajectory characteristics of this detection, it is not suitable for linear fitting (not satisfied with the premise of monotonically increasing or decreasing). However, if we discard the data segment in the red rectangular boxes and calculate along the zonal direction (this is also the way to use linear fitting for this case, and the fitting direction is the black dashed line in figure R5a), we can recalculate the K(1) value to 0.004. This shows that some of the original detection trajectories are too irregular in the process of zonal or

meridional projection and can not satisfy the decomposition of this direction.

However, we have added to the revised version of the manuscript the results of the calculation after fitting a straight line along the trajectory (leaving out the irregular curved trajectory). Both these and the results of the zonal/meridional decomposition are presented in the manuscript. One reason is for comparison of the results containing different physical flow regions and the results containing only a single physical flow region, and the other reason is to explore whether the presence of strong wind field fluctuations (before and after linear fitting) will affect the correlation between the calculated small-scale gravity wave parameters and other disturbance parameters.

Specifically, what criteria is used for reference to decompose according to the longitude direction or the latitude direction?

We take the direction of the longer projection distance as the separation distance direction.

Specifically, if the line from the start point to the end point of the flat-floating segment is longer in the projection along the zonal (meridional) direction, the zonal (meridional) direction is chosen as the separation distance direction.

Can the data analysis performed after the decomposition still represent the characteristics of the fluctuation?

Since the wind field always comes from a combination of two directions (meridional and zonal), our selection of one direction as the separation distance direction actually reflects only part of the characteristics of the wave. Of course, the best way is to adopt the method of linear fitting, but the data segment of linear fitting is limited, and the overly curved trajectory cannot be covered, and this part of the content contains important fluctuation information.

Therefore, in order to take into account the region of strong fluctuations and the fluctuation information of the synthesized wind direction, we give the results of taking the latitude/longitude direction as the separation distance direction and the fitting straight line as the separation distance direction in the manuscript.

5) L154-L161: Why was the gravity wave scale chosen to be 5km? Considering the different horizontal resolutions of different data, the specific scale of gravity waves selected should be different. Therefore, I understand that the author here should choose the wave parameter closest to 5km, and suggest that the author give the statistical distribution results of the actual scale.

In addition, considering that the flat-floating distance is long enough, why not choose a longer scale, and whether this will cause a difference in the analysis results?

**Response**: Sorry for this gross oversight, but the threshold set in our program is 6km.

The scale of gravity waves comes from a separation distance r less than 6 km and closest to 6km. The statistical characteristics of H1 and C1 at the corresponding scale were used as the quantization of perturbation parameters of small-scale gravity waves at this scale, which was applied in earlier studies.

we make the following modifications:

L159 Changed "and the separation distance closest to 5 km (< 5 km)"

To "and the separation distance closest to 6 km (< 6 km)".

In fact, choosing the scale closest to 6km can not only satisfy the statistical quantity of

parameter results, but also ensure the robust of velocity increment on this scale. We now take the data from Wuhan on October 23 as an example to explain this choice.

[Figure]

**Figure R7.** Velocity increments calculated from beginning to end in the data series where the separation distances are (a) 64 m, (b) 512 m, (c) 4096 m, and (d) 32 768 m, respectively.

Since the structure function is a statistical feature and we only focus on small-scale GWs (within a few kilometers) and turbulence, the deviation of the structure function which may be caused by the obvious change of the floating height on a larger scale (the influence of floating height on the statistical characteristics of velocity increment gradually increases, as shown in Fig. R7) is not within the scope of our discussion. Therefore, the results are reasonable and reliable.

With the increase of the selected scale, the set of data points of the speed increment du decreases, and its fluctuation value becomes more and more distinguishable. That is, too long a scale will cause significant differences in the velocity increments du, and the result will be no longer robust and cannot be used to calculate H1 and C1.

At the same time, we supplement the statistical results of the selected gravity wave scale, shown in figure R8.

[Figure]

**Figure R8.** Histogram of gravity wave scale.

6) L167-L169: How does the author determine this conclusion? It is suggested that the author add relevant explanations (figures or tables) in this work.

**Response**: Thanks for your comments, we make the following modifications:

Changed "The velocity increments δu_L (r) is the key process for calculating the disturbance parameters from flat-floating data, and has shown good robustness within the separation distance of small-scale gravity waves and turbulence (He et al., 2022a). thereforeTherefore, the results will not be affected by the fluctuation of flat-floating height, as well as the swing of the balloon."

to "The velocity increments $\delta u_L(r)$ is the key process for calculating the disturbance parameters from flat-floating data, and has shown good robustness within the separation distance of small-scale gravity waves (Figure A2). In fact, choosing the scale closet to 6 km (less than 6 km) can not only satisfy the statistical quantity of parameter results, but also ensure the robust of velocity increments on this scale. With the increase of the separation distance, the fluctuation of velocity increments becomes more and more distinguishable. That is, too long a scale will cause significant differences in the velocity increments du at different data points, and the result will be no longer robust and cannot be used to calculate H1 and C1. Therefore, the selected SGW scale of 6 km will not be affected by the fluctuation of flat-floating height, as well as the swing of the balloon."

[Figure]

**Figure R9.** Velocity increments calculated from beginning to end in the data series from Yichang site on November 8, where the separation distances are (a) 44 m, (b) 352 m, (c) 5600 m, and (d) 45 056 m, respectively.

7) From Figure 2, according to my understanding, the Hurst parameter H1 and intermittency parameter C1 both come from the calculation of the slope. If so, the slope of H1 is easily understood, which comes from the linear fitting of the structure function spectrum in Figure 2a. However, how is the slope of C1 calculated in Figure 2d?

Also, how to determine the premise of Eq. (9)? Given a nonstationary random atmospheric process with stationary increments that is scale invariant from some outer scale R down to some inner scale η, I wonder whether the relationship K(1)=0 can always be satisfied when calculating the intermittent parameter C1? In other words, how is approximately close to 0 (L166) defined, and does it eliminate part of the result?.

**Response**: In response to the first question, we make the following modifications:

L175 Changed "Figure 2c is the relationship between the $q$-order singularity measure $\langle \varepsilon(r;x)^q \rangle$ and the separation distance $r$ in log-log coordinate, through which C1 is calculated with a value of 0.08 (Figure 2d)."

To "Figure 2c is the relationship between the $q$-order singularity measure $\langle \varepsilon(r;x)^q \rangle$ and the separation distance $r$ in log-log coordinate. The curves $q=1$, $q=2$, $q=3$, $q=4$, and $q=5$ are given, from which the slope values can be calculated within the selected SGW scale (left of the black dashed line) as -$K(1)$, -$K(2)$, -$K(3)$, -$K(4)$, and -$K(5)$, respectively. Then the variation curve of K(q) with q can be obtained in Figure 2d, where $q=0$, 0.25, 0.5, …, 5. The fitting slope of the K(q) curve at q=1 is calculated from the K(q) values

corresponding to q=0.75, q=1, and q=1.25, and this slope vale is defined as intermittent parameter C1."

In response to the second question, allow me to explain the problem from two aspects:

a. Based on mathematical formula.

Based on the mathematical formula 9, to obtain the value of C1, the numerator and denominator must both approach 0 as q approaches 1, thus obtaining the value of C1 according to the L'Hôpital's rule. The derivation of C1 is given in detail in Marshak et al, 1997.

b. Analysis based on specific cases.

Let's take WH case at October 18 pm as an example to illustrate it. In this example, the calculated K(1) is 0.04.

The result of the flat-floating trajectory is shown in Figure R5. This is projected in the meridional direction. It should be noted that the two red rectangles in Figure R5a are dominated by the zonal displacement. In Figure R5c, the periods with small Y displacement over time are the periods between 0-593s and 9111-11500s, respectively.

We also plot the change of zonal and meridional winds over time, as shown in Figure R6. In this case, the longitudinal velocity ($u_L$) is the meridional wind component (v). According to the formula 5-9, the larger value of K(1) is mainly due to the larger ensemble average value of the velocity increment (that is, the ensemble average of the longitudinal velocity difference over the separation distance r on the whole data).

In Figure R6, the longitudinal velocity increment in the two red rectangular boxes (the same time period as in Figure R5) has several segments for the sudden increase velocity, and the presence of these segments causes the $\delta u_L$ (r) value to be larger.

Due to the trajectory characteristics of this detection, it is not suitable for zonal decomposition (not satisfied with the premise of monotonically increasing or decreasing along the zonal). However, if we discard the data segment in the red rectangular boxes and calculate along the zonal direction (this is also the way to use line fitting for this case, and the fitting direction is the black dashed line in figure R5a), we can recalculate the K(1) value to 0.004. This shows that some of the original detection trajectories are too irregular in the process of zonal or meridional projection and can not satisfy the decomposition of this direction.

[Figure]

**Figure R10.** slope K(q) obtained (a) before and (b) after linear fitting.

When considering all physical flow regions (as mentioned in our previous reply), K(1)

serves as a threshold to exclude the conditions that do not satisfy the statistical characteristics of the intermittent parameters in this case. Our definition of K(1) approximately close to 0 in this manuscript means that K(1)<0.01.

When K(1) exceeds this value, we consider that the data series does not satisfy an unstable process with stable increments. It can also be intuitively seen from the K(q) curve that K(1) and 0 have a certain distance. This means that the trajectory is too curved and irregular, or the actual detection of wind speed has too much wild value (from the positioning system of excessive error).

we make the following modifications:

L166 Added "Here $K(1)$ approximately close to 0 is defined as $K(1)$<0.01. When $K(1)$ exceeds this value, it can be intuitively seen from the $K(q)$ curve that $K(1)$ and 0 have a certain distance. The physical explanation behind it is that the flat-floating trajectory is too irregular, or the actual detected wind speed has too many wild values (abnormalities from the positioning data)."

8) L173-L175: "From third-order structure function, a downscale energy cascade (from large to small scales) can be seen from the third-order structure function".

How do you know this conclusion? Do red dots (negative values) represent downscale energy cascade, corresponding to a drag of gravity wave on background, while blue dots (positive values) denote upscale energy cascade corresponding to an increase in turbulent kinetic energy? I think the authors should use more approachable language that explains its physical meaning.

**Response**: Yes, red dots (negative values) represent downscale energy cascade, corresponding to a drag of gravity wave on background, while blue dots (positive values) denote upscale energy cascade corresponding to an increase in turbulent kinetic energy.

Based on Kolmogorov theory, Lindborg (Lindborg, 1999) obtained the ideal theoretical relationship for two-dimensional turbulence by reprocessing. Experiments show that the third-order structure function is superior to the second-order structure function in analyzing turbulence (Cho & Lindborg, 2001; Lindborg & Cho, 2001). The third-order structure function can not only eliminate the arbitrariness of the universal constant in the power law expressed by the second-order structure function in physical space, but also reflect the direction of the energy cascade through the sign of the value, where negative values represent downscale energy cascades and positive values represent inverse scale energy cascades. And this conclusion is obtained from above researches.

The direction of the energy cascade can be obtained from equation 1. Since $E$ is the energy dissipation rate, when $S_3(r)$ is positive, $E$ is negative, meaning that energy is transferred from small to large scales. when $S_3(r)$ is negative, $E$ is positive, meaning that energy is transferred from large to small scales.

Thanks for your comments, we make the following modifications:

L121 Changed "A positive value of $S3(r)$ represents upscale energy cascades (from small to large scales), while a negative value of $S3(r)$ represents downscale energy cascades (from large to small scales) (Lindborg, 1999)."

To "The relationship between the third-order structure function and the energy transfer can be obtained by Eq. (1). When $S_3(r)$ is positive, $E$ is negative, energy transfers from small to

large scales, meaning upscale energy cascades. When $S_3(r)$ is negative, $E$ is positive, energy transfers from large to small scales, meaning downscale energy cascades."

9) L180: The curve in Figure 2d increases as q increases. Could you make more explanations on the physical meaning of K(q)? What does the larger or smaller K(q) (i.e., intermittent parameter) mean?

**Response**: In fact, K(q) is not an intermittent parameter, K(q) is the fitting slope of $\left\langle \varepsilon(r)^q \right\rangle$ and r in the log-log coordinate system, where q is the order. The intermittent parameter is the slope of the tangent of the curve of K(q) at q=1. The intermittency parameter is defined to measure the singularity of a fluctuation. The more singular a fluctuation, the larger the intermittency and the fewer the information dimensions, resulting in the fluctuation being more like the Dirac delta function. With this interpretation, the intermittency can be understood as an information codimension of the fractality of a process.

The value of the intermittency parameter is typically in the range of $0 \leq C1 = K'(1) \leq 1$.

The larger C1 is, the more obvious the characteristic of turbulent activity is. In general, 0.2-0.3 can already represent significant turbulence. In the current investigation results, the value of C1 in the turbulence occurrence data is also in this range. (plasma physics study by Carreras et al. (2000) came up with values for intermittency in plasma turbulence ranging from 0.15 to 0.30,

10) l90-192: This statistical result is very interesting, and it is recommended that the author supplement its physical explanation.

**Response**: In our previous discussion, it has been shown that when the flat-floating trajectory is significantly curved (the trajectory is not straight), it indicates that there are different physical flow regions in the detection region, and when this part of the curved trajectory is not omitted, bidirectional energy cascades can often be found in the obtained third-order structure function, which means the instability of the wind field fluctuation.

When the value of H1 is smaller, the instability of the data series is stronger. In the trajectory diagram drawn, the flat-floating trajectories of the six sites in autumn are more irregular, indicating more frequent different physical flow regions, and internal instability is more likely to occur.

Thanks for your comments, we make the following modifications:

L192 Added "It is reasonable to have a lower H1 distribution in autumn, since the flat-floating trajectories of the six sites in autumn are more irregular. The obvious change in the trajectory (away from the previous straight direction) indicates that the detected data contains different physical flow regions, suggesting internal instability of the background wind field fluctuations and some multifractal characterizations."

11) Explanation of the jagged structure in the spectral shape.

By comparing the results of Figure 2 and Figure 3, we can see some differences in the results:

a) Compare (a) in Figure 2 with (a) and (e) in Figure 3, multi-order structure function has

obvious spectral shape difference on large scale (larger than 10km). What is the difference behind this difference in the actual observed wind field? It is suggested that the authors add relevant explanations.

b) Compare (b) in Figure 2 with (b) and (f) in Figure 3, third-order structure function has obvious the sawtooth structure in the spectral shape in Figure 3 (b) and (f), while the spectral shape in Figure 2 (b) is much smoother. Can sawtooth be considered a special spectral structure that appears when atmospheric disturbances are strong?

**Response**: To illustrate the first problem, we plot the longitudinal velocity component and flat-floating trajectory of the three cases (stable gravity wave, unstable gravity wave, and the coexistence of gravity wave and turbulence) as shown in Figure R11. The obvious spectral shape difference on large scale (larger than 10 km) mainly comes from the intervals with significant inclinations accompanied by a relatively large increase or decrease in the speed increment $u_L(r)$ on these intervals. Since $S_q(r) = \langle |\delta u_L(r)|^q \rangle$, when the curve of multi-order structure function at a certain separation distance r has an obvious inflection point, it means that there is a sudden increase or decrease of some velocity increment in the set of all velocity increment at this scale.

[Figure]

**Figure R11.** The variation of horizontal velocity component uL along the zonal separation distance (left panel) and flat-floating trajectory (right panel) from three cases.

To illustrate the second problem, we observe the trajectories of three cases. For the stable gravity wave (Yichang site on November 8), the flat-floating trajectory moves approximately along a quasi-straight line, reflecting a relatively single physical flow region, which indicates that the internal instability of atmospheric wind field fluctuations is relatively weak. For the unstable gravity wave (Yichang site at October 15 pm) and the coexistence of gravity waves and turbulence (Yichang site at November 10 pm), the flatfloating trajectory has been significantly deflected, indicating that the detection area contains different physical flow regions, which means that the internal instability of atmospheric wind field fluctuations is relatively strong. Obviously, this also caused the sawtooth structure in the spectral shape and the inconsistency in the energy cascade direction of the third-order structure function.

Thanks for your comments, we make the following modifications:

L187 Added "By comparing the case results of Figure 2 and Figure 3, multi-order structure function (third-order structure function) can be found to have the spectral shape differences on certain scales, which mainly comes from the intervals with significant inclinations accompanied by a relatively large increase or decrease in the speed increment $u_L(r)$ on these intervals (Figure A3). Since $S_q(r) = \langle |\delta u_L(r)|^q \rangle$, when the curve of $S_q(r)$ at a certain separation distance r has an obvious inflection point, it means that there is a sudden increase or decrease of some velocity increment in the set of all velocity increment at this scale (He et al., 2022a).

For the stable gravity wave (Yichang site on November 8), the flat-floating trajectory moves approximately along a quasi-straight line (Figure A3b), reflecting a relatively single physical flow region, which indicates that the internal instability of atmospheric wind field fluctuations is relatively weak. For the unstable gravity wave (Yichang site at October 15 pm) and the coexistence of gravity waves and turbulence (Yichang site at November 10 pm), the flat-floating trajectory has been significantly deflected (Figure A3d and A3f), indicating that the detection area contains different physical flow regions, which means that the internal instability of atmospheric wind field fluctuations is relatively strong. Obviously, this also caused the sawtooth structure in the spectral shape and the inconsistency in the energy cascade direction of the third-order structure function.
"

12) L199-L206: I think the author's method here for calculating inertial gravity waves and turbulence is too simplistic. The author is strongly advised to further elaborate on the details. Taking into account the specific nature of the journal, detailed and complete discussion is encouraged.

For example, what is the calculated altitude interval of the turbulence parameter? Are the statistical results from the regional average or the regional sum?

In addition, considering that there are already many observations of inertial gravity waves and turbulence, it is recommended to increase the comparison between the relevant parameters of this article and the existing results to further illustrate the rationality and reliability of the results.

**Response**: Thanks for your comments, we fully agree with your suggestion and make the following modifications:

a) The expression of method introduction is expanded and the introduction to methods is moved to section 3.

Added "Since turbulence is highly intermittent, the turbulence parameters obtained here are derived from the regional average of non-zero values (turbulence exists) within the height range of 15-25 km of each profile."

b) The comparison with the IGW and turbulence parameter results of other studies is

supplemented.

L217 Added "In this paper, the vertical wavelength of the IGW is concentrated in the range of 1~3 km, which is close to the scale of the stratospheric IGW in China (1.5~3 km) observed by radiosonde data (Bai et al., 2016). In our results, kinetic energy and potential energy of IGW are concentrated at 2~6 J/kg and 0~2 J/kg, respectively. In the tropics, by contrast, the kinetic energy of stratospheric IGW has already exceeded 10 J/kg (Nath et al., 2009), indicating more intense wave activity at lower latitudes. The turbulent kinetic energy dissipation rate $log_{10}\varepsilon$ is between -5 and -2 from RTISS, which is comparable to those obtained based on radiosonde data in the United States from $-4$ to $-0.5$ $m^2 s^{-3}$ (Ko and Chun, 2022) and in Guam from $-6$ to $0$ $m^2 s^{-3}$ (He et al., 2020a). "

13) L255: In Figure 7, the authors illustrate the correlation between the different parameters by drawing a scatter plot. However, due to the limited sample size, some linear relationships are not obvious, which makes some of the author's statements seem a little absolute. For example, L267-L270: The trend of increasing first and then decreasing is not obvious, so it is suggested that the author delete this expression.

Also, If the author tends to discuss the relationship between inertial gravity waves and small-scale gravity waves, the relatively absolute wording is modified to a mild expression. Because in the present work of the authors (if the number of current observations cannot be significantly increased), due to the limitations of the sample, it is also necessary to discuss the possibility that the maximum or minimum value of the edge region in the scatter results is caused by the wild value.

**Response**: Thanks for your comments, we fully agree with your suggestion and make the following modifications:

a) Deleted the place where the expression of relevance is not obvious.

Deleted "The turbulent kinetic energy dissipation rate ($\varepsilon$) increases first and then decreases with the increase of KHI (Figure 7h). This is because the increase of KHI is conducive to the generation of turbulence, however, when the KHI reaches a certain threshold value, the turbulent layer cannot be maintained and begins to decay, resulting in a weakening of turbulence activity (He et al., 2020b)."

b) To discuss the possibility that the maximum or minimum value of the edge region in the scatter results is caused by the wild value.

We recalculated H1 and C1, where quality control and pre-judgment were performed for each selected flat-floating segment. Considering that the calculation of wind speed comes from the coordinates of the positioning system, the pre-judgment is to observe the difference between longitude and latitude at adjacent times. If the curve has no outlier value, it indicates that the positioning system works normally in the flat drift stage, and the obtained wind speed is also credible.

The judgment method is shown in Figure R12, left panel shows the case where the positioning data is abnormal, and right panel shows the case where the positioning data is normal.

[Figure]

**Figure R12.** (a) wind velocity, (c) latitude difference, and (e) longitude difference for the case where the positioning data is abnormal, and (b) wind velocity, (d) latitude difference, and (f) longitude difference for the case where the positioning data is normal

Obviously, the difference of positioning coordinates in adjacent time can identify the abnormal situation of positioning data, that is, there are a large number of wild values in a stable increment. Even if there is a sudden increase in wind speed, the transformation of the positioning data should be continuous, and this wild value comes from the anomaly of the signal received by the positioning system. So the data in this case is discarded.

Under this premise, the presence of larger C1 values can completely eliminate the interference of outlier values. Although the sample size of a large C1 value (>0.15) is relatively small (because strong detection of intermittent parameters is inherently a small probability event in the total sample), it is still possible to see that a large C1 does correspond to a weakened inertial gravity wave below.

According to your suggestion, we make the following modifications:

L195 Added "Considering that the calculation of wind speed comes from the coordinates of the positioning system, it is necessary to make sure that there is no wild value interfering with the results. The difference of positioning coordinates in adjacent time can identify the abnormal situation of positioning data, that is, weather there are obvious wild values in the difference of longitude or latitude. Figure A4 shows the cases for abnormal and normal positioning data, and these abnormal cases are screened out."

L270 Added "Although the quantity of large C1 values (>0.15) is relatively rare (the detected disturbances with strong intermittence are still small probability events in the entire sample), it is still possible to see that the enhanced C1 is accompanied by the weakened IGW below."

14) L282-L283: "Based on the ERA5 reanalysis data, the ozone mass mixing ratio (OMR) and PV at different pressure layers that matched the detection are selected"

How this is matched needs to be further explained.

**Response**: Thanks for your comments, we fully agree with your suggestion and make the following modifications:

Changed "Based on the ERA5 reanalysis data, the ozone mass mixing ratio (OMR) and PV at different pressure layers that matched the detection are selected. According to the latitude and longitude range covered by RTISS during flat-floating stage, the OMR and PV obtained from the ERA reanalysis data are averaged in the corresponding area"

To "Based on the ERA5 reanalysis data, the ozone mass mixing ratio (OMR) and PV at different pressure layers that matched the detection are selected. Specifically, the ERA5 data at 00UTC and 12UTC within the longitude and latitude range of the selected flat-floating stage are screened, and the value after regional average is used as the reanalysis data result corresponding to the flat-floating detection at that time."

15) L354: What exactly do fingerprints mean here?

**Response**: What the author wants to express here is that the parameter space (H1 and C1) is used to related to changes in atmospheric composition indirectly. Considering that the expression may not be clear, we make the following modifications according to your suggestion:

Changed "Besides, the possible "fingerprint" of this parameter space in the material transport of other components (such as water vapor, carbon dioxide, methane, etc) in the stratosphere also deserves further attention."

To "Besides, potential connections that may exist between this parameter space and other atmospheric components (such as water vapor, carbon dioxide, methane, etc) transported in the stratosphere also deserves further attention."

Minor comments:

1) L65: The two colors in Figure 1 are not marked with seasons, please add.

**Response**: Thanks for your comments, according to your suggestion we have made the following modifications:

The figure has been redrawn to add a description of the colors.

2) L168: Where is Text S2?

**Response**: I am very sorry that this is an earlier version of the expression and it is not in the current manuscript. It has been deleted.

3) L200: 18–25km → 18–25 km.

**Response**: Thank you for pointing out this detail, and we have checked and corrected similar problems in the whole article.

4) L213: "critical layer filtering" should be further explained, for example, how are gravity waves affected here by the background wind field.

**Response**: Thanks for your comments, according to your suggestion we have made the

following modifications:

Changed "The dominant propagation directions of IGWs in summer and autumn are northeast and southwest respectively, due to the effect of "critical layer filtering" (Eckermann, 1995)."

To "The dominant propagation directions of IGWs in summer and autumn are northeast and southwest respectively, due to the effect of "critical layer filtering" (Eckermann, 1995). The background wind field filters out gravity waves propagating in the same direction, and passes through gravity waves propagating in the opposite direction."

5) L220: The display in figure4 e and f is incomplete and the authors should readjust the boundary values.

**Response**: Thanks for your comments, according to your suggestion we have made the following modifications:

The figure is redrawn and the longitudinal scale ranges of the subgraphs e and f are adjusted.

6) L220: The two colors in Figure 1 are not marked with seasons, please add.

**Response**: Thanks for your comments, according to your suggestion we have made the following modifications:

The figure has been redrawn to add a description of the colors.

7) L235: The size of the ordinate scale in Figure 5b is inconsistent with other subgrams.

**Response**: Thanks for your comments, according to your suggestion we have made the following modifications:

The figure has been redrawn to adjust the smaller scale value in the vertical coordinate of the subgraph.

8) L253: between  H1 → between H1

**Response**: Thanks for your comments, according to your suggestion we have made the following modifications:

Changed "between  H1"

To "between H1"

9) L321-L324 100hPa → 100 hPa, 10hPa → 10 hPa

Please check for similar errors elsewhere.

**Response**: Thanks for your comments, according to your suggestion we have made the following modifications:

Changed "100hPa" to "100 hPa"

To "10hPa" to "10 hPa"

At the same time, we examined the entire manuscript and corrected for the occurrence of multiple spaces between words.

9) Unfortunately, neither my abilities nor my time allow me to find all the grammatical problems throughout the manuscript. Therefore, I sincerely ask the authors to check the full text by themselves, and preferably seek advice from a native speaker.

**Response**: Thank you for pointing this out. We have revised the whole manuscript once again, including the grammatical problems. I hope you could understand that there has been no specific language polishing given that the current version of the manuscript may still need to be revised. If you still feel that the language needs further improvement in the future, we will increase the language polishing in the subsequent revision.

At the end, Authors are grateful to the anonymous reviewer for providing valuable comments to improve the manuscript up to this level. We greatly appreciate the time and effort you put into improving the quality of my manuscript, and we have benefited immensely from your selfless comments and suggestions. Besides, if you have more suggestions or comments about my manuscript or the content of the reply, I will always be pleased to make timely replies and revisions and benefit from communicating with you. Finally, thank you again from the bottom of my heart.

**In addition, the author also checked the full text, revised some grammar and details, and they can all be found with "track changes".**

---

## Author Response (AR2)

The editorial support team
*Atmospheric Chemistry and Physics*
January 15th, 2023
Subject: Revision of manuscript egusphere-2023-1608

Dear Editors

Thank you for your letter and for giving us the opportunity to revise our manuscript on "Identification of stratospheric disturbance information in China based on round-trip intelligent sounding system" [Paper # egusphere-2023-1608]. We have carefully reviewed the comments and have revised the manuscript accordingly. Our responses are given in a point-by-point manner below. Changes to the manuscript are shown in the revised manuscript with "track changes".

Sincerely,

Yang He

E-mail: heyang12357@sina.com

Corresponding author: Zheng Sheng
E-mail: 19994035@sina.com

**Response to Reviewer #1:**

General comments:

The authors answered the comments carefully and used the discussion to improve the manuscript from both the scientific and technical point of view. Even though some argumentation seems to be a bit weak to me (see section Specific comments), it might be just a question of personal perspective and the answers generally give impression that the methods were verified. I fully understand that processing of observations must contain some approximations and that the authors have to work with limited amount of data. However, the manuscript still needs to be checked for technical problems, some of which are listed in the section Technical corrections.

**Response**: Thank you for your understanding and recognition of our revised work. Without your help and advice, the manuscript would not have been significantly improved. Thank you again for your time and effort in the evaluation of our work. We have carefully reviewed the comments and have revised the manuscript accordingly. Our responses are given in a point-by-point manner below. Changes to the manuscript are shown in the revised manuscript with "track changes".

Specific comments:

1) If the results/time series properties should be, due to the smoothing, used only for the internal comparison in the manuscript, I think this should be mentioned in the text (at least I did not find it there).

**Response**: Thank you for pointing out this detail, and according to your suggestion, we have added this expression in the corresponding part of the manuscript:

L114:

Added "It should be noted that different smoothing points may cause some difference in

the quantization results of SGWs. However, if all data sets are smoothed in the same way, the internal comparison will not be affected by this."

2) I appreciate the effort put to the testing of the different (fitted) separation direction. However, it is not completely what I meant. I do not really see reason to subset the data after doing the linear regression. Of course, the data are not linear so the direction would not follow the trajectory for all the times. I understood the fitted direction just as a natural direction of the x axis. From my point of view, it was just to ensure that there will be no methodology difference between the trajectories that head mostly in the zonal direction and the trajectories heading in other directions. And when taking the zonal separation direction, the trajectories parts that change to another direction are also not removed. Or is there some methodology problem I am missing? Considering this, I cannot agree with the argumentation that the linear fitting method does not lead to the correct result since only part of the data can be used. On the other hand, I like the supplemented information about the physical flow regions.

**Response:**

Thank you very much for your understanding and recognition of my supplementary work. At the same time, your understanding is very correct. We mainly treat the fitted direction just as a natural direction of the x/y axis to process the original trajectory, and retain the trajectories parts that change to another direction. This processing method can include more detection of multi-physical flow regions. When these trajectories (the wind direction changes greatly over time) are fitted linearly, it is difficult to satisfy the term "fitting", that is, the fitted straight line fits the actual trajectory ideally. Based on this, we propose a single physical flow region, that is, we selected examples with a relatively single direction of the trajectory, and perform linear fitting along the trajectory direction.

Therefore, the linear fitting method can actually get more accurate results. However, the problem is that because many curved trajectories are rounded out after screening, the results obtained are not suitable for internal comparison. And irregularly curved trajectories may also contain important disturbance information. Compared with the best linear fitting of the single-physical flow region, the zonal or meridional projection in the multi-physical flow region can be said to be a compromise method. Not only can more samples be retained, but also the disturbance information behind the curved/irregular trajectories can be retained.

Thank you very much for your valuable comments. If you could understand that I would like to keep both methods, but focus more on the comparison of the results of the multi-physical flow regions, I would be very grateful. Of course, regarding the analysis of the single physical flow region (a more ideal situation), we will continue to follow up in future research. When the number of samples is large enough, even if all fits are unified in one direction, there are still a sufficient number of perfectly fitted samples (the fitted straight line is ideally in line with the actual trajectory), which can ensure the robustness of the statistical results.

According to your suggestion, we make the following changes to the expression:

Changed "After this treatment (linear fitting), the omitted part may correspond to the large fluctuation region of the wind field, which will also cause the loss of atmospheric disturbance information"

To "If the trajectory direction is relatively single (single-physical flow region), the linear fitting method can actually get more accurate results. However, the problem is that because many curved trajectories are rounded out after screening, the results obtained are not suitable for internal comparison. And irregularly curved trajectories may also contain important disturbance information. Compared with the best linear fitting of the single-physical flow region, the zonal or meridional projection in the multi-physical flow zone can be said to be a compromise method. Not only can more samples be retained, but also the disturbance information behind the curved/irregular trajectories can be retained."

3) I am confused about the Figure A5: If it shows statistically significant relationship, why not show it in the main text instead of the insignificant results?

**Response**: Because in Figure 9, the selected Ek+Ep represents the total energy of the inertial gravity wave, the purpose is to first give the reader a preliminary intuitive understanding of the trend of the total energy with C1. This selection is similar to the momentum flux that includes the zonal and meridian directions, which is aimed to first show whether there is a connection between the total energy characteristics of IGW and the C1 changes of SGW. It is equivalent to a logic from overall to local characteristics, so Figure 9 is shown first, and Figure A5 is discussed later.

Technical corrections:
1) L46: "Stephen A et al., 2015" – check if the citation is correct.
**Response**: Thank you for pointing out this detail, we have corrected the citation information.
Changed "Stephen A et al., 2015" to "Cohn et al., 2013"

2) L82: Remove quotation mark.
**Response**: According to your suggestion, we have made corresponding modifications.

3) L96: "The three-stage detection process by RITSS described in Figure 2." – missing verb.
**Response**: According to your suggestion, we have made corresponding modifications:
Changed "The three-stage detection process by RITSS described in Figure 2" to "The three-stage detection process by RITSS is described in Figure 2".

4) L137, L155, L165, L167, L170, L172: Please correct signs after equations.
**Response**: According to your suggestion, we have made corresponding modifications.

5) L148, L153, L163: Capital letter in the middle of sentence.
**Response**: According to your suggestion, we have made corresponding modifications.

6) L142, 157: Missing space before bracket.

**Response**: According to your suggestion, we have made corresponding modifications.

7) L156: Two commas.

**Response**: According to your suggestion, we have made corresponding modifications.

8) e.g., L269: Some plot captions end with dot, some not.

**Response**: We uniformly corrected it to all without dots at the end.

9) L287: Units should not be in italic.

**Response**: We checked the entire manuscript and corrected all the units to non-italics.

10) L325: Two dots.

**Response**: According to your suggestion, we have made corresponding modifications.

11) L426: Equation 1 should be capitalised.

**Response**: According to your suggestion, we have made corresponding modifications. Changed "equation 1" to "Eq. 1".

12) L436: Section 5 should start on a new line.

**Response**: According to your suggestion, we have made corresponding modifications.

13) L463: "regardless of whether it has been linearly fitted or not" The object "it" is not so clear, please reformulate.

**Response**: According to your suggestion, we have made corresponding modifications. Changed "it" to "the flat-floating trajectory".

At the end, Authors are grateful to the anonymous reviewer for providing valuable comments to improve the manuscript up to this level. We greatly appreciate the time and effort you put into improving the quality of my manuscript, and we have benefited immensely from your selfless comments and suggestions. **Besides, if you have more suggestions or comments about my manuscript or the content of the reply, I will always be pleased to make timely replies and revisions and benefit from communicating with you.** Finally, thank you again from the bottom of my heart.